# Superior colliculus bidirectionally modulates choice activity in frontal cortex

Alyse Thomas[1,3], Weiguo Yang[1,3], Catherine Wang [1], Sri Laasya Tipparaju[1], Guang Chen[1], Brennan Sullivan [1], Kylie Swiekatowski[1], Mahima Tatam[1], Charles Gerfen [2] & Nuo Li [1] ✉

Action selection occurs through competition between potential choice options. Neural correlates of choice competition are observed across frontal cortex and downstream superior colliculus (SC) during decision-making, yet how these regions interact to mediate choice competition remains unresolved. Here we report that SC can bidirectionally modulate choice competition and drive choice activity in frontal cortex. In the mouse, topographically matched regions of frontal cortex and SC formed a descending motor pathway for directional licking and a re-entrant loop via the thalamus. During decision-making, distinct neuronal populations in both frontal cortex and SC encoded opposing lick directions and exhibited competitive interactions. SC GABAergic neurons encoded ipsilateral choice and locally inhibited glutamatergic neurons that encoded contralateral choice. Activating or suppressing these cell types could bidirectionally drive choice activity in frontal cortex. These results thus identify SC as a major locus to modulate choice competition within the broader action selection network.

In our moment-to-moment activities, we must choose appropriate actions and disregard alternatives. Models of decision-making and action selection suggest that appropriate actions are selected through competition between potential choice options[1,2]. These models generally invoke pools of neurons representing competing choice options, and actions are selected through competition between these neuronal populations. A neural correlate of choice competition has been observed across frontal cortex[3–5], basal ganglia[6,7], as well as downstream superior colliculus (SC)[8–15], where distinct neuronal populations encode potential choice options and exhibit push-pull dynamics reflecting choice competition. While this choice activity is widespread and likely reflects a distributed process across the broader action selection network[16–18], the circuit mechanism for choice competition remains poorly understood.

Traditional models of actions selection suggest that the frontal cortex and basal ganglia mediate action selection[19–23]. SC receives outputs from these regions[24–29]. Recent empirical evidence has also established a role for SC in decision-making and action

selection[8–15,30–41]. Modulating frontal cortex[35,42–44] or SC[12,15,32–37,40,42] on their own can bias choice behavior, but the relationship between these regions and how they implement choice competition remain to be determined. Traditional models of action selection implicate SC to be downstream to the frontal cortex and suggest that SC biases selection processes after the frontal cortex[19–22]. Other works suggest the two regions work cooperatively[15,16,42]. Neurophysiological recordings comparing the frontal cortex and SC found that choice activity arises simultaneously in both regions[16], or even earlier in SC in some cases[15]. Few studies have causally probed the interactions of these regions during action selection.

A frontal cortical region in the mouse, anterior lateral motor cortex (ALM), is critical for planning and initiation of directional licking[5,43,45–47]. We mapped activity in topographically connected regions of ALM and SC to identify activities responsible for the selection and initiation of directional licking in mice. Within topographically matched regions of ALM and SC, we identified intermingled neuronal populations in both regions that represent contralateral and ipsilateral

[1]Department of Neuroscience, Baylor College of Medicine, Houston, TX, USA. [2]Section on Neuroanatomy, National Institute of Mental Health, Bethesda, MD, USA. [3]These authors contributed equally: Alyse Thomas, Weiguo Yang. ✉e-mail: nuol@bcm.edu

choice options and exhibit competitive dynamics prior to licking response. By manipulating SC and examining the impact on the choice activity, we found that modulating SC could bidirectionally and persistently bias choice activity in ALM during action selection. Cell-type specific recordings and manipulations in SC further revealed a circuit mechanism for choice competition: within each SC hemisphere, GABAergic neurons encode ipsilateral choice and inhibit glutamatergic neurons encoding contralateral choice, and the two SC hemispheres mutually inhibit each other, which collectively drive competitive interactions between contra-preferring and ipsi-preferring populations in ALM.

## Results

### Topographically matched regions in ALM and SC mediate selection and initiation of directional licking

Small amounts of anterograde tracers injected into the medial and lateral regions of ALM revealed topographical projections in the central and lateral portions of SC respectively (Fig. 1a, b and Supplementary Fig. 1a–e). The lateral ALM overlapped with the orofacial representation of motor cortex where unilateral stimulation triggers contralateral licking[5,48,49], also known as the tongue-jaw motor cortex[50]. Its projection target, the lateral SC, is also thought to control licking behaviors in mice[14,25,33]. Anterograde tracer injections in the lateral SC revealed projections that overlap with descending ALM projections to the contralateral medulla (Fig. 1b and Supplementary Fig. 1f-h), including the intermediate nucleus of the reticular formation (IRt) presynaptic to the hypoglossal nucleus that drives the tongue[5]. Retrograde labeling from the medulla showed that IRt-projecting SC neurons were concentrated in the lateral region of SC (Fig. 1c and Supplementary Fig. 1i), coinciding with the lateral ALM projection zone (Fig. 1b and Supplementary Fig. 1e). Thus, the lateral ALM and lateral SC formed a descending motor pathway (Fig. 1a).

In addition, SC sends ascending projections back to ALM via the thalamus[14,51–53]. We combined anterograde and retrograde tracer injections to map connectivity between SC and ALM in the thalamus (Fig. 1d). SC projections overlapped with the ALM-projecting thalamus, including parts of the ventral-medial nucleus, medial-dorsal nucleus, and parafascicular nucleus (Fig. 1d and Supplementary Fig. 1j-l). Retrograde tracer injections in thalamic nucleus VM labeled neurons across the lateral and central regions of SC (Fig. 1c and Supplementary Fig. 1i), encompassing the medial ALM projection zone (Fig. 1b and Supplementary Fig. 1e). Thus, the medial ALM and the central SC formed a re-entrant loop (Fig. 1a). Together, these anatomical data identified topographically matched regions of ALM and SC that form a descending motor pathway to IRt and an ascending cortico-collicular loop via the thalamus[53,54].

To examine SC's role in directional licking, we activated or inhibited SC during a delayed response task, in which mice discriminated object location during a sample epoch then reported choice using directional licking following a delay epoch (Fig. 1e, Methods). Mice performed the task with high performance and few early licks. In mice expressing channelrhodopsin-2 (ChR2) in SC glutamatergic output neurons (Vglut2-ires-cre mice or wildtype mice injected with ChR2 viruses, Methods), unilateral photostimulation of SC prior to the response epoch evoked premature licking to the contralateral direction ('SC photoactivation', Fig. 1f). Unilateral SC photoactivation evoked contralateral licking even when mice were not engaged in the delayed response task (Supplementary Fig. 2a, b). This is consistent with SC's descending projections to the contralateral medulla (Fig. 1b), which arise from SC glutamatergic neurons[14]. To test if SC glutamatergic output was required for the initiation of directional licking in the delayed response task, we bilaterally photostimulated SC GABAergic neurons in VGAT-ChR2-EYFP mice (Methods), which inhibited the glutamatergic neurons ('SC photoinhibition'). Bilateral SC photoinhibition during the response epoch blocked licking responses

(Fig. 1g and Supplementary Fig. 2c). These results show that the lateral region of SC is involved in initiating directional licking.

Next, we examined SC contributions to action selection by manipulating SC during specific sub-epochs of the task prior to the response epoch. Unilateral SC photoinhibition biased licking to the ipsilateral direction relative to the manipulated hemisphere, resulting in lower performance in trials where mice were instructed to lick to the contralateral direction and improved performance in the ipsilateral licking trials (Fig. 1h and Supplementary Fig. 2d-f). Photoinhibition began to bias future choice during the late sample epoch and the effect size grew progressively stronger during the delay epoch (Fig. 1h). SC receives long-range inhibitory projections from other brain regions including the basal ganglia substantia nigra pars reticulata[25,27,53], which could be activated by photostimulation in VGAT-ChR2-EYFP mice. However, photostimulating ChR2 locally expressed in SC GABAergic neurons (in GAD2-ires-cre mice) also induced the same pattern of bias (Supplementary Fig. 2j), indicating that the bias resulted from inhibition of SC. The effect was spatially restricted to the lateral region of SC as photoinhibition in the medial region of SC induced little lick direction bias (Supplementary Fig. 2k).

In contrast to SC photoinhibition, photoactivation of SC in Vglut2-ires-cre x Ai32 mice during the sample and delay epochs biased future licking to the contralateral direction (Fig. 1i and Supplementary Fig. 2g-i). To activate SC without evoking premature licking, we used lower photostimulation power (0.1-0.2 mW, Methods; Supplementary Fig. 2i). SC photoactivation biased future lick direction starting from the sample epoch and the effect size grew stronger during the delay epoch (Fig. 1i). Neither photoactivation nor photoinhibition significantly altered licking execution or reaction time (Supplementary Fig. 2e, f, h, i). These results show that transiently activating or suppressing SC could bias future choice long after the photostimulation, consistent with a role of SC in the selection of directional licking.

To identify the activity responsible for action selection and movement initiation, we mapped task selectivity in topographically matched regions of ALM and SC using silicon probes (Fig. 2a). We labeled the recording locations using fluorescence dye and aligned the recorded neurons into the Allen Mouse Common Coordinate Framework (Methods, Fig. 2b; ALM, 51 penetrations in 22 mice, 2939 neurons; SC, 57 penetrations in 16 mice, 1147 neurons). We tested for significant trial-type selectivity of individual neurons using spike counts during the sample and delay epochs (three-way ANOVA, Methods). By using both correct trials and error trials, trial types differed by the instructed tactile stimulus (anterior for "lick left" vs. posterior for "lick right") or choice (licking left vs. licking right) (Fig. 2c). A population of neurons exhibit trial-type selectivity as defined by the tactile stimulus across both correct and error trials (Fig. 2d and Supplementary Fig. 3c, 'stimulus-selective'; ALM: 336/2468; SC: 41/621; $p < 0.01$ for stimulus, three-way ANOVA). Stimulus selectivity emerged early in both ALM and SC during the sample epoch (Supplementary Fig. 3d, i), with the strongest selectivity observed after stimulus onset. Another population of neurons showed ramping selectivity for lick direction before the licking response (Fig. 2d and Supplementary Fig. 3e, 'choice-selective neurons'; ALM: 464/2468; SC: 70/621; $p < 0.01$ for choice, three-way ANOVA). Choice selectivity emerged simultaneously in ALM and SC during the late sample epoch and was the strongest during the delay (Supplementary Fig. 3f, j). Separately, we identified neurons based on significant firing rate modulation during the licking response. A population of neurons show activity phase-locked to rhythmic lick cycles (Fig. 2d and Supplementary Fig. 3g, 'licking movement neurons'; ALM: 208/2939; SC: 264/1147; Methods) and the majority of these neurons were active before the first lick (Supplementary Fig. 3h, k), consistent with a motor command.

Spatially, our electrophysiological mappings revealed a medial-lateral gradient of activity reflecting a progression from choice to motor command, which unfolded across topographically matched

regions of ALM and SC (Fig. 2e, f). The stimulus- and choice-selective neurons were concentrated medially in ALM and SC (Fig. 2e, f), coinciding with the medial ALM projection zone in SC that in turn projects back to ALM via the thalamus (Fig. 2g, h). The licking movement neurons were enriched laterally in ALM and SC (Fig. 2e, f), coinciding with the lateral ALM projection zone in SC that in turn projects to the IRt (Fig. 2g-i). Thus, the medial region of ALM and central region of SC were reciprocally connected through the thalamus and these regions

were enriched in choice selectivity. The lateral regions of ALM and SC formed a descending pathway to the medulla which were enriched with licking movement activity.

### Push-pull dynamics between neuronal populations encoding competing choices

To examine activity dynamics mediating the selection of directional licking, we next focused on topographically matched regions of ALM

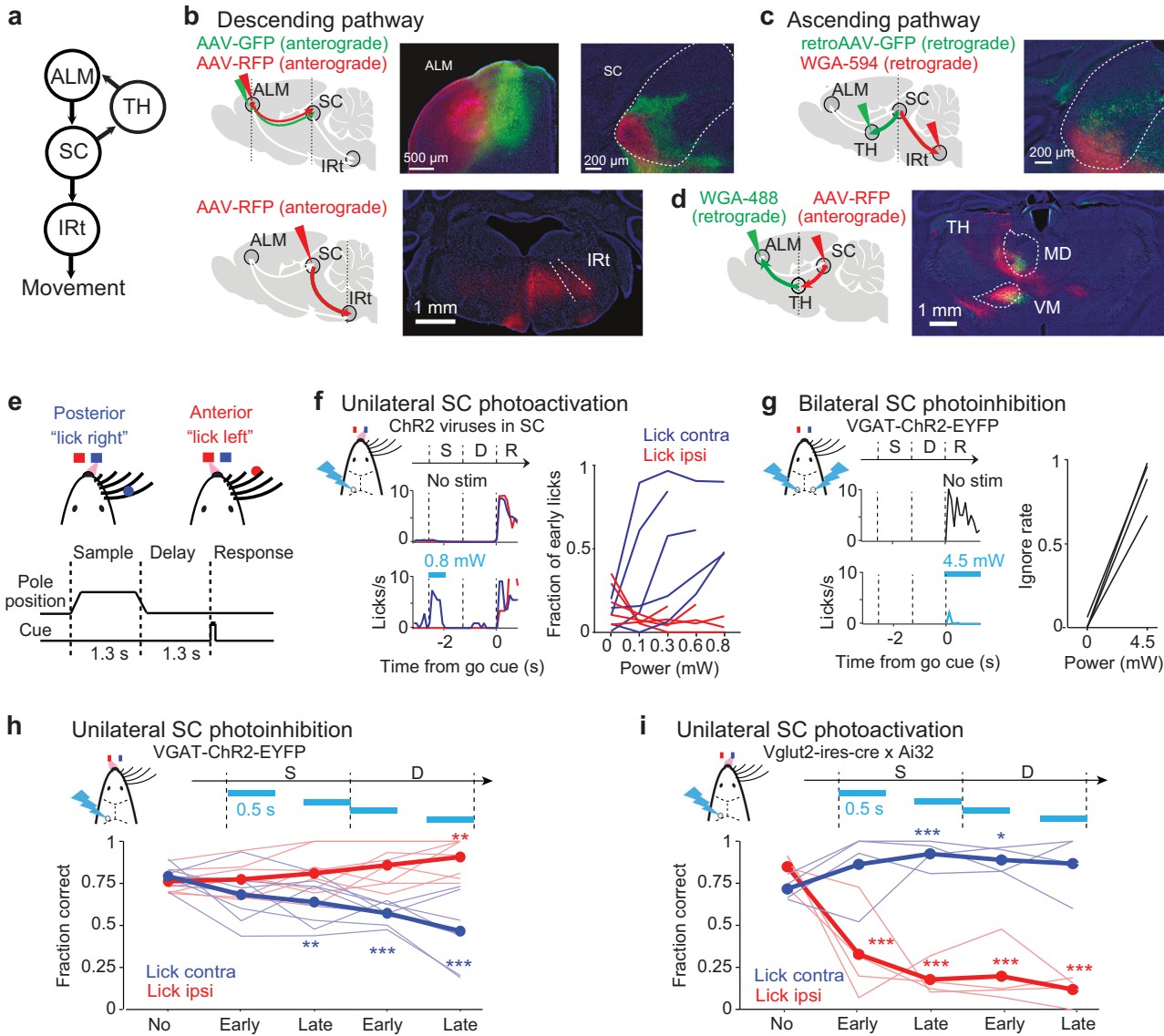

**Fig. 1 | A topographical ALM-SC circuit mediates initiation and selection of directional licking. a** Schematic of the ALM-SC circuit. **b** Labelling descending projections. *Top*, Organization of ALM-SC projections. Two-colored anterograde tracer injections in the medial (green) and lateral ALM (red) and their projections in SC (*n* = 4 animals). *Bottom*, anterograde tracer injection in the lateral SC and its projections in IRt (*n* = 3 animals). Borders of SC and IRt are based on Allen Reference Atlas. White lines show unlabeled parts of the region. **c** Organization of SC thalamus-projecting (green) neurons and IRt-projecting neurons (red). Two-colored retrograde tracer injections in VM and IRt (*n* = 3 animals). White lines show unlabeled parts of the region. **d** Ascending SC projections (red) and ALM-projecting thalamus (green). Borders of VM and MD are based on Allen Reference Atlas. *N* = 3-4 animals per group. **e** *Top*, head-fixed mouse performing behavioral task. *Bottom*, whisker detection occurs during the sample epoch. After a delay period, an auditory 'go' cue signals the response epoch. **f** Unilateral SC photoactivation triggers contralateral licking. *Left*, average lick rate from an example mouse. Blue,

contralateral licking relative to the manipulated hemisphere; red, ipsilateral licking. Dashed lines, behavioral epochs. Cyan, photostimulation. *Right*, fraction of 'early lick' trials as a function of laser power. Individual lines, individual mice (*n* = 5). Color indicates direction of the early lick. **g** Bilateral SC photoinhibition blocks licking. *Left*, average lick rate from an example mouse. *Right*, fraction of trials with no lick response as a function of laser power. Individual lines, individual mice (*n* = 4). **h** Performance with transient unilateral SC photoinhibition. Performance is the fraction of correct choices, excluding lick early trials and no lick trials. Thick lines, mean; thin lines, individual mice (*n* = 8). \**p* < 0.05; \*\**p* < 0.01; \*\*\**p* < 0.001; one-tailed t-test, bootstrap (Methods). Trials are grouped by instructed lick direction relative to the manipulated hemisphere. Blue, contralateral (lick contra); red, ipsilateral (lick ipsi). Photostimulation power, 1.2 mW. **i** Performance with transient unilateral SC photoactivation. Thick lines, mean; thin lines, individual mice. \* *p* < 0.05; \*\**p* < 0.01; \*\*\**p* < 0.001; one-tailed t-test, bootstrap (Methods). *N* = 4 mice. Photostimulation power, 0.1–0.2 mW.

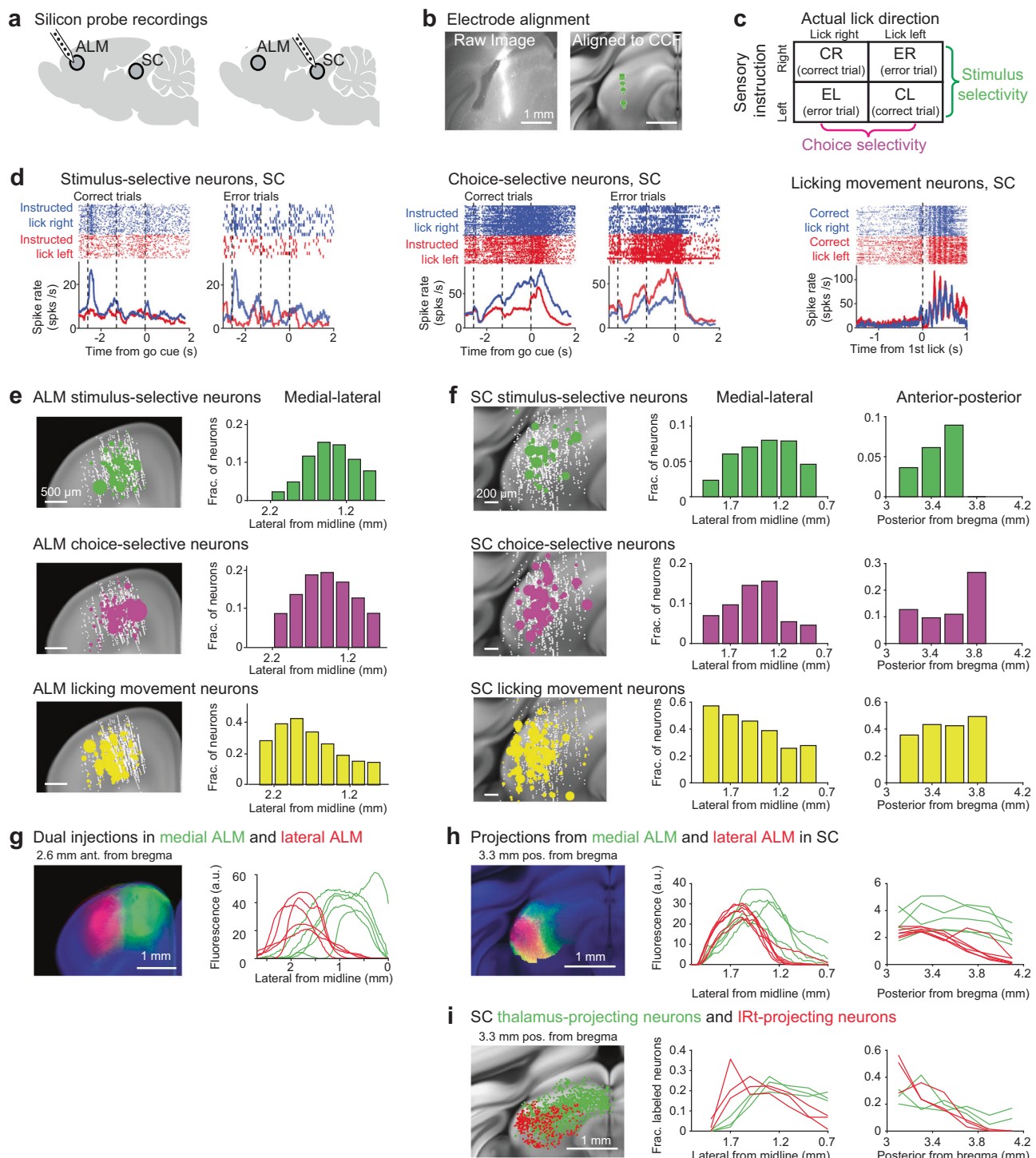

and SC enriched with choice selectivity, i.e., the medial region of ALM and central region of SC (Fig. 3a, the regions highlighted by green tracer injection). We examined how neuronal populations encoded "lick left" and "lick right" choices by calculating selectivity of individual neurons as the firing rate difference between "lick left" and "lick right" trials in specific epochs of the task (Fig. 3a and Supplementary Fig. 4a-d; Methods). For this analysis, only correct trials were used as many neurons were recorded for a limited number of error trials. Prior to the response epoch, neurons with significant selectivity emerged in both ALM and SC, and distinct neuronal populations showed preferences for either contralateral or ipsilateral choices (Fig. 3a, sample and delay epochs). In both regions, contra-preferring and ipsi-preferring neurons were spatially intermingled and present in roughly equal proportions

during the delay epoch (Fig. 3b; proportion of contra- vs. ipsi-preferring neurons, ALM, $p = 0.42$; SC, $p = 0.29$, bootstrap test). This contrasted with the selectivity during the response epoch, which exhibited a contralateral preference in SC (Fig. 3b; $p < 0.001$, bootstrap test). We found that many ipsi-preferring SC neurons during the sample and delay epochs switched their choice preferences to encode contralateral lick direction during the response epoch (Supplementary Fig. 4d)[14]. Distinct choice preferences of these SC neurons across behavioral epochs suggest they may play distinct roles in action selection vs. movement initiation.

We considered the possibility that choice selective activity during the delay epoch might be attributable to mice's ongoing movements, which could differ in "lick left" and "lick right" trials. To address this, we

**Fig. 2 | Topographically organized ALM-SC circuits show a gradient of stimulus, choice, and movement activity. a** *Left*, silicon probe recordings in ALM and SC. **b** Recording tracks labeled and aligned into the CCF (Methods). An example recording track in SC with single units (green dots, *n* = 8 units). **c** Calculating selectivity using correct and error trials. Significant selectivity is tested with three-way ANOVA (Methods). **d** Example neurons in SC selective for stimulus, choice, and movement (Methods). For stimulus- and choice-selective neurons, raster and peristimulus time histograms (PSTHs) are shown for correct and error trials. Licking movement neurons show spike rate modulation during rhythmic licking cycles. **e** *Left*, spatial map of ALM stimulus-selective neurons (*n* = 336), choice-selective neurons (*n* = 464), and licking movement neurons (*n* = 208). Dot size indicates the strength of selectivity stimulus- and choice-selective neurons and spike rate modulation for licking movement neurons. White dots show all recorded neurons. *Right*, medial-lateral (ML) distribution of selective neurons. The fraction of neurons is relative to all recorded neurons within each bin. The distribution of stimulus-selective neurons differs significantly from licking movement neurons (*p* < 0.001,

permutation test, bootstrap, Methods); the distribution of choice-selective neurons differs significantly from licking movement neurons (*p* < 0.05, bootstrap). **f** Same as **b** for SC stimulus-selective neurons (*n* = 41), choice-selective neurons (*n* = 70), and licking movement neurons (*n* = 264). The ML distribution of stimulus-selective neurons differs significantly from licking movement neurons (*p* < 0.01, permutation test, bootstrap); the distribution of choice-selective neurons differs significantly from licking movement neurons (*p* < 0.01, bootstrap). **g** Labelling of ALM projection neurons. *Left*, two-colored anterograde tracer injections to map descending projections. *Right*, fluorescence profiles along the ML axis. Individual lines reflect individual cases (*n* = 6). **h** Localization of ALM projection zones in SC. *Left*, coronal section showing projections from the medial (green) and lateral (red) ALM. *Right*, fluorescence profiles along the ML axis and anterior-posterior (AP) axis. Lines, individual injection cases (*n* = 6). **i.** Localization of SC projection neurons. *Left*, coronal section showing SC thalamus-projecting neurons (green) and IRt-projecting neurons (red). *Right*, distributions of SC thalamus-projecting neurons and IRt-projecting neurons. Lines, individual injection cases (*n* = 3).

built convolutional neural networks (CNN) to predict neurons' firing rate from videos of orofacial movements (Methods). The model predicted a significant portion of ALM and SC activity on single trials (Supplementary Fig. 4e-g). We then subtracted this movement-related activity from ALM and SC activity and the choice selectivity remained in the residuals (Supplementary Fig. 4h). This video analysis shows that ongoing movements could not explain the choice selectivity during the delay epoch. Moreover, the gradual buildup of the choice selectivity over the delay epoch closely mirrored the effect size of SC photoinhibition, which induced the strongest bias in upcoming choice during the delay epoch (Fig. 1h), consistent with a role of this choice activity in action selection.

We next examined the dynamics of contra-preferring and ipsi-preferring populations prior to the licking response. In ALM, our silicon probes permitted simultaneous recording of the two populations (i.e., neurons sorted by their lick direction preference, 12 neurons in each population on average, 17 sessions from 9 mice). For each recording session, we averaged the activity of all the neurons in each population and examined their dynamics in single trials (Methods). The activity of contra- and ipsi-preferring populations was anticorrelated across the time course of a single trial: when the activity of contra-preferring population fluctuated upward, the activity of ipsi-preferring population fluctuated downward (Fig. 3c). This push-pull was highly dynamic in single trials, sometimes even flipping signs during the sample and delay epochs (Fig. 3c and Supplementary Fig 5a), and the state of this push-pull at the end of the delay epoch predicted mice's choice (Fig. 3d and Supplementary Fig. 5a-b). To quantify this push-pull dynamic across time, we calculated Pearson's correlation between the activity of contra- and ipsi-preferring populations on single trials. Robust anticorrelation could be observed at the level of single-trial activity (Fig. 3e; lick contra trials, r = −0.27 ± 0.01; lick ipsi trials, r = −0.27 ± 0.01; mean ± s.e.m. across trials).

The anticorrelated dynamics were not observed if neurons were randomly grouped into two populations (Fig. 3e and Supplementary Fig. 5b; r = 0.06 ± 0.01). Moreover, the anticorrelation was absent before the sample epoch (lick contra trials, r = 0.05 ± 0.02; lick ipsi trials, r = 0.00 ± 0.02) and was diminished during the response epoch (lick contra trials, r = 0.03 ± 0.02; lick ipsi trials, r = 0.01 ± 0.02). In addition to their anticorrelated activity across time within each trial, the activity states of contra- and ipsi-preferring populations were also anticorrelated across trials, i.e. 'noise correlation'. We calculated noise correlation between the two populations within the same trial type (within "lick left" or "lick right" trials), using mean-subtracted activity at the end of the delay epoch to examine their co-fluctuations across trials (Supplementary Fig. 5c). We observed a significant negative noise correlation between contra- and ipsi-preferring populations across trials, consistent with a push-pull interaction between the two populations (Supplementary Fig. 5d).

In SC, recordings yielded fewer simultaneously recorded neurons due to the small volume of SC regions that are connected with ALM (9 neurons in each population on average, 9 sessions from 5 mice). Nevertheless, similar anticorrelated activity between contra- and ipsi-preferring populations was also observed (Supplementary Fig. 5e-f). The effect was restricted to the central region of SC where choice-selective neurons were enriched (Supplementary Fig. 5f).

Together, these analyses revealed push-pull dynamics between neuronal populations encoding contralateral and ipsilateral choices prior to the licking response, which suggests choice competition that gives rise to the selection of directional licking.

## SC can bidirectionally drive choice competition dynamics

What circuit mechanism mediates the competition between choice representations? Our optogenetic experiments showed that activating or inhibiting SC during the delay epoch could bidirectionally bias upcoming choice (Fig. 1h, i), suggesting that SC contributed to the selection of upcoming choice. However, the relationship between ALM and SC and their respective roles in choice competition remained unclear. Choice competition could occur in ALM and SC could reflect the output from ALM to bias downstream processes. Alternatively, SC could influence choice competition in ALM through its ascending projections.

To determine the influence of SC on choice competition, we performed electrophysiological recordings in ALM and SC while unilaterally photoinhibiting SC in VGAT-ChR2-EYFP mice. We first examined the direct effect of SC photoinhibition on contra-preferring and ipsi-preferring populations within SC by performing optrode recordings (Fig. 4a). For contra-preferring population, SC photoinhibition suppressed the population activity on average (Fig. 4a). Interestingly, activity was selectively suppressed in trials where mice were instructed to lick contralaterally relative to the manipulated hemisphere ('lick contra trials') whereas the activity was little affected in lick ipsi trials (Fig. 4a and Supplementary Fig. 6a). For ipsi-preferring population, SC photoinhibition enhanced the activity on average and mainly in lick contra trials (Fig. 4a and Supplementary Fig. 6a). Thus SC photoinhibition caused opposing activity changes in contra- and ipsi-preferring populations, which recapitulated their competitive interactions during normal choice competition. Because this biasing of activity mainly occurred in lick contra trials, with little activity change in lick ipsi trials, SC photoinhibition thereby rendered the overall activity pattern to mimic that of the lick ipsi trials, concordant with the ipsilateral choice bias in behavior (Fig. 1h).

We next examined the impact of SC manipulation on ALM choice activity. In ALM, unilateral SC photoinhibition similarly induced an opposing activity change in a trial-type dependent manner. The average activity of contra-preferring population was selectively suppressed in lick contra trials (Fig. 4b and Supplementary Fig. 6b). At the same

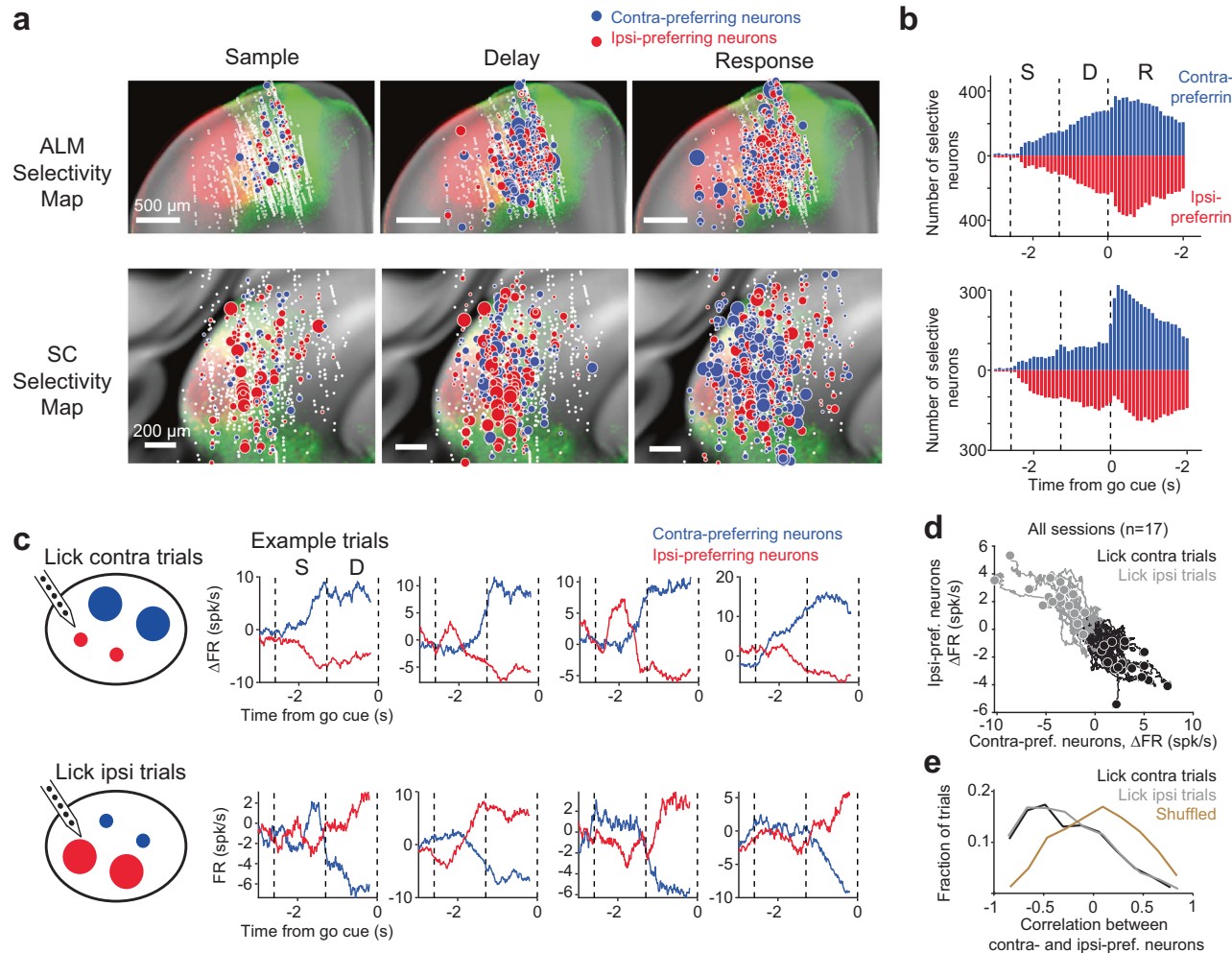

**Fig. 3 | Neurons in ALM and SC encode contralateral and ipsilateral choices and show competitive dynamics. a** Spatial map of contra-preferring (blue circles) and ipsi-preferring neurons (red circles) in ALM (*top* row) and SC (*bottom* row) during specific behavioral epochs (ALM, 3.36-2.27 mm anterior from bregma in CCF, 2468 neurons, 22 mice; SC, 2.90-3.91 mm posterior from bregma, 621 neurons, 16 mice). Dot size indicates the size of the selectivity during specific epochs. Selectivity is the firing rate difference between preferred and non-preferred trial type. White dots show all recorded neurons. Selectivity maps are overlaid onto fluorescence showing descending ALM-SC projections from the medial ALM (green) and lateral ALM (red). **b** Number of significantly selective ALM (*top*) and SC neurons (*bottom*) as a function of time. Significant selectivity is based on spike counts in 200-ms time windows, $p < 0.01$, two-tailed t-test. Neurons are sorted by their lick direction preference (blue, contra-preferring; red, ipsi-preferring). Dashed lines, behavioral epochs. **c** *Left*, cartoon depicts the competitive dynamics between contra-

preferring and ipsi-preferring neurons across trial types. *Right*, activity (ΔFR) of ALM contra-preferring (blue) and ipsi-preferring (red) neurons in single trials. Data from an example session. Activity reflects firing rate change from the mean where the mean firing rate across trial types is subtracted to yield ΔFR (Methods). *Top row*, lick contra trials (lick direction is relative to the recorded hemisphere). *Bottom row*, lick ipsi trials. **d** Activity (ΔFR) of ALM contra-preferring versus ipsi-preferring neurons. Individual lines show average ΔFR of individual sessions. Dots show ΔFR at the end of the delay epoch. Data from 9 mice and 17 sessions. Black, lick contra trials; gray, lick ipsi trials. **e** Pearson's correlation between ΔFR of contra-preferring and ipsi-preferring neurons in single trials. Correlation is calculated using ΔFR during the sample and delay epochs. Data from 1606 trials. Black, lick contra trials. Gray, lick ipsi trials. Yellow, shuffled control where neurons are randomly grouped into two populations without regard to their choice preference. Lick contra and lick ipsi trials vs. shuffled control, $p < 0.001$, two-tailed t-test.

time, the activity of ipsi-preferring population was enhanced in lick contra trials (Fig. 4b and Supplementary Fig. 6b), recapitulating their push-pull with the contra-preferring population during choice competition. Activity in lick contra trials was thus rendered to be similar to lick ipsi activity, consistent with the ipsilateral choice bias in behavior (Fig. 1i). Notably, the average activity of contra-preferring and ipsi-preferring populations changed little in lick ipsi trials (Supplementary Fig. 6b). In lick ipsi trials, SC photoinhibition induced a mixture of excitation and inhibition within both populations (Supplementary Fig. 6d, e). As a result, the average activity of each population was unchanged because excitation and inhibition canceled out each other. This balanced activity change in lick ipsi trials contrasted sharply with the push-pull dynamics induced by SC photoinhibition in lick contra trials (Supplementary Fig. 6d, e).

To quantitively test whether SC photoinhibition induced a trial-type specific biasing of choice activity rather than a non-specific activity change, we calculated the activity change between control and photostimulation trials for each neuron in each trial type (Supplementary Fig. 6a, b). We then modeled the activity changes of contra-preferring and ipsi-preferring populations with linear models. The model has a term to capture non-trial-type-specific activity changes and a term to capture trial-type-dependent activity changes ($\beta_0$ and $\beta_1$, respectively; Methods). We additionally added a random effect of recording sessions so any session-specific activity changes are absorbed by the random effect parameters and only activity changes common across sessions will be captured by the model. In both SC and ALM, the model consistently showed a trial-type-dependent activity change: SC photoinhibition induced a significant activity decrease of

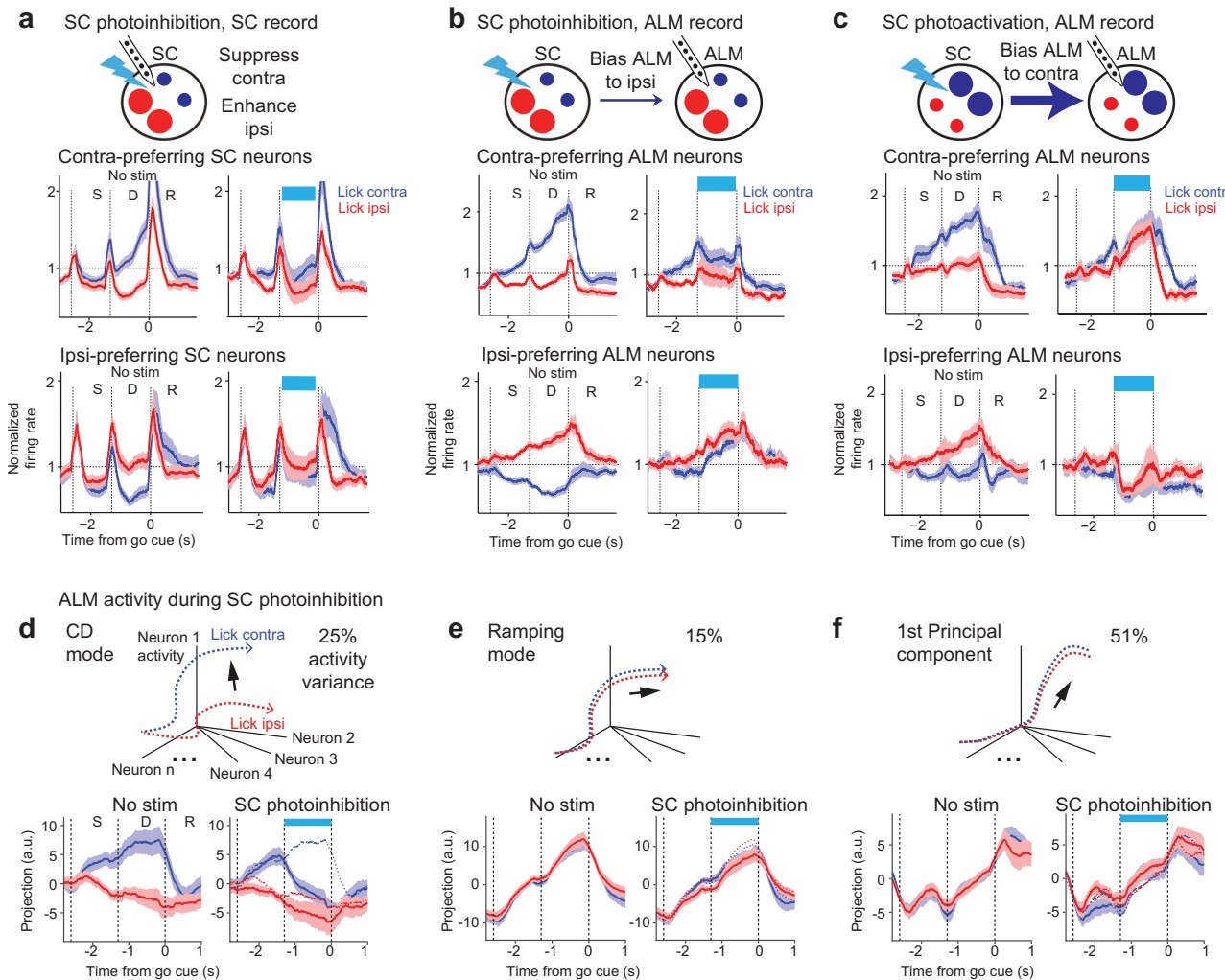

**Fig. 4 | SC bidirectionally drives push-pull dynamics between competing choice representations. a** *Top*, schematic of SC recording during SC photoinhibition. *Middle*, comparison of activity in contra-preferring SC neurons during control (*left*) and photostimulation (*right*). *Bottom*, activity in ipsi-preferring SC neurons. Population response of SC neurons in lick contra trials (blue) and lick ipsi trials (red). Both correct and error trials are included, grouped by instructed lick direction relative to the recorded hemisphere. Only neurons with significant trial-type selectivity during the delay epoch are included. Neurons are grouped by their preferred trial type using spike counts from 10 trials and the remaining data is used to compute the population response. The spike rate of each neuron is normalized to the mean spike rate across all trial types. Activities of contra-preferring and ipsi-preferring neurons are first averaged within each session and the plots show mean

± s.e.m. across sessions (34 sessions, 15 mice). **b** Same as **a** but for ALM recording during SC photoinhibition in the same hemisphere. $N = 40$ sessions, 12 mice. **c**. Same as **a** but for ALM recording during SC photoactivation in the same hemisphere. $N = 10$ sessions, 4 mice. **d** *Top*, schematic of choice-specific activity trajectories and coding dimension (***CD***) in activity space. *Bottom*, ALM activity in control and SC photoinhibition trials projected onto the ***CD***. Mean ± s.e.m. (bootstrap, Methods). $N = 33$ sessions, 10 mice. **e** ALM activity along a ramping direction that captures non-selective ramping activity during the delay. Line, mean. Shading, SEM. **f** ALM activity along the principal component that captures the most activity variance. Activity variance is quantified over time across both sample and delay epochs. Line, mean. Shading, SEM.

contra-preferring population specifically in lick contra trials (Supplementary Fig. 6a-b; SC: nontrial-type specific change $\beta_0$, $p = 0.85$, trial-type specific change $\beta_1$, $p < 0.01$; ALM: $p = 0.91$ and $p < 0.001$, respectively), whereas SC photoinhibition induced a significant activity increase of ipsi-preferring population specifically in lick contra trials (SC: non-trial-type specific change $\beta_0$, $p = 0.93$, trial-type specific change $\beta_0$, $p < 0.05$; ALM: $p = 0.98$ and $p < 0.001$, respectively).

To test if SC could bidirectionally modulate choice competition dynamics in ALM, we photoactivated SC in Vglut2-ires-cre x Ai32 mice while recording from ALM (Fig. 4c). SC photoactivation during the delay epoch biased ALM activity to contralateral choice dynamics in the stimulated hemisphere, opposite to the activity change induced by SC photoinhibition. Contra-preferring neurons were enhanced and ipsi-preferring neurons were depressed, again recapitulating their push-pull dynamics during choice competition. These changes

occurred selectively in lick ipsi trials (Supplementary Fig. 6c), and they biased overall activity towards contralateral choice dynamics (Fig. 4c).

SC manipulations induced a highly specific biasing of ALM choice activity rather than a global change of population activity. To illustrate this feature of the activity change, we analyzed ALM activity in an activity space where individual dimensions corresponded to the activity of individual neurons. We decomposed activity into several orthogonal modes (Methods). We first projected the population activity onto a coding dimension (***CD***) in activity space along which activity maximally differentiated "lick left" and "lick right" choice during the delay epoch (Fig. 4d). Our previous analyses showed that ALM activity along the ***CD*** was tightly coupled to behavioral choice[47,55,56]. During SC photoinhibition, ALM activity was selectively biased along the ***CD***, where the activity trajectory in lick contra trials was collapsed to lick ipsi activity trajectory (Fig. 4d). We additionally

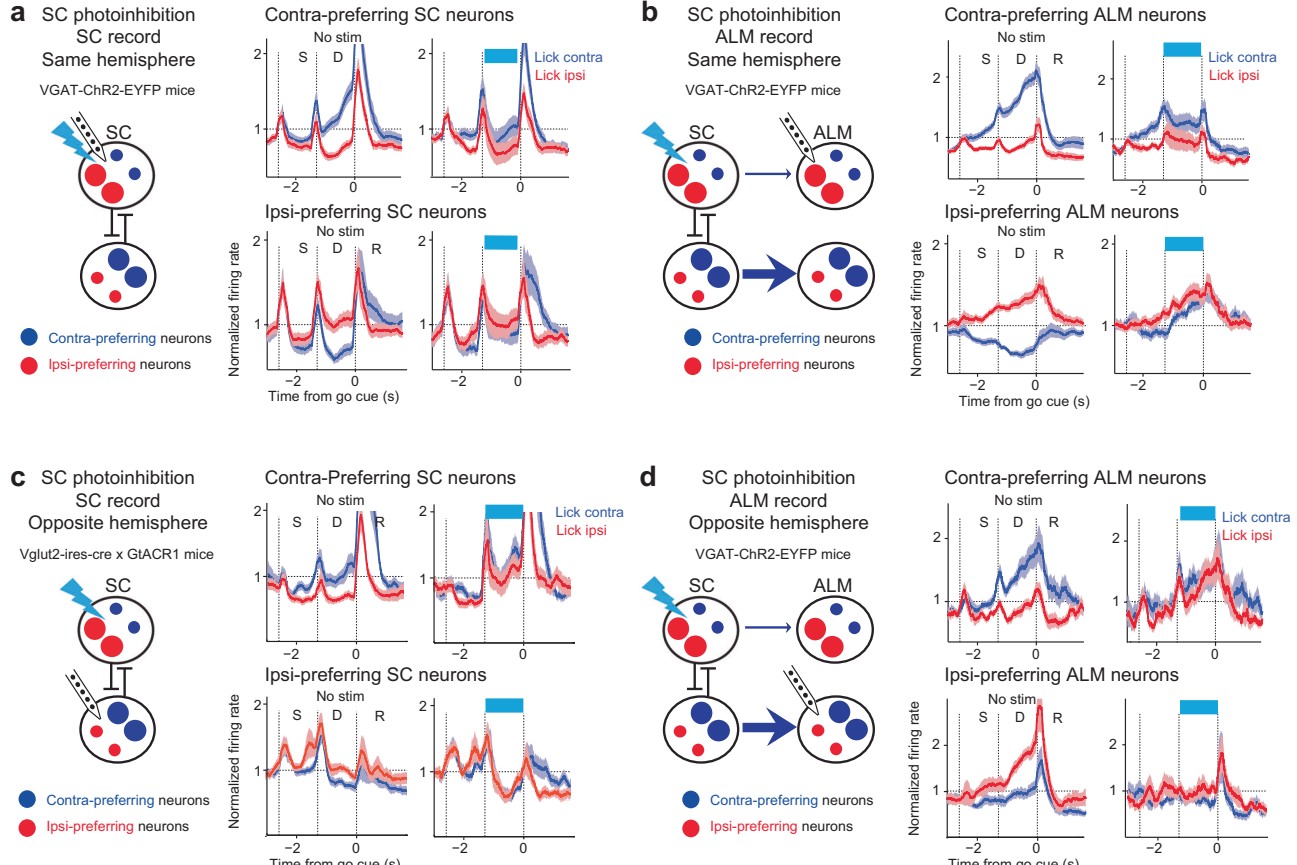

**Fig. 5 | SC photoinhibition biases choice representations across both hemispheres. a** Unilateral SC photoinhibition during SC recording in the same hemisphere. Blue, lick contra trials; red, lick ipsi trials. Data from Fig. 4a replotted here for comparison. **b** Unilateral SC photoinhibition during ALM recording in the same hemisphere. Data from Fig. 4b replotted here for comparison. **c** Unilateral SC photoinhibition during SC recording in the opposite hemisphere. $N = 6$ sessions, 3 mice. Line, mean. Shading, SEM. **d** Unilateral SC photoinhibition during ALM recording in the opposite hemisphere. $N = 11$ sessions, 6 mice. Line, mean. Shading, SEM. In both SC and ALM, SC photoinhibition in the opposite hemisphere enhances activity of contra-preferring population and suppresses activity of ipsi-preferring neurons.

examined two other modes of ALM delay activity. One activity mode captured a non-trial-type selective ramping activity that has been associated with animals' reaction time (Fig. 4e)[47,55–58]. SC photoinhibition minimally affected ALM ramping activity (Fig. 4e). Finally, we obtained the first principal component of ALM population activity that captured the majority of activity variance (~51%) and found little change in activity during SC photoinhibition (Fig. 4f). Together, these analyses revealed a highly specific effect of SC on ALM choice activity.

Unilateral photoinhibition of SC drove coordinated changes in choice activity across both hemispheres. Recordings in both SC and ALM of the opposite hemisphere showed that whereas the activity pattern in the stimulated hemisphere was biased to ipsilateral choice dynamics, the activity pattern in the other hemisphere was simultaneously biased to contralateral choice dynamics (Fig. 5), i.e., the opposite pattern of activity change from the stimulated hemisphere. Similar to the manipulated hemisphere, unilateral SC photoinhibition biased choice competition in the other hemisphere by oppositely modulating the activity of contra- and ipsi-preferring neurons (Fig. 5c-d), recapitulating their competitive dynamics during normal choice competition. This suggests that the two SC hemispheres act in a coordinated and antagonistic fashion, consistent with interhemispheric competition[33,59,60]. Thus, each hemisphere of SC promoted contralateral choice dynamics in the same hemisphere while suppressing contralateral choice dynamics in the opposite hemisphere.

Together, these results show that suppressing or activating SC could bidirectionally drive competitive interactions between ALM populations encoding opposing choice options, positioning SC as a potential locus to bias the competition between choice representations.

## Transient perturbation reveals that SC can persistently bias choice activity in ALM

ALM and SC are reciprocally connected (Fig. 1a-d), and earlier works show that modulating ALM could also bias choice activity and choice behavior[5,44,55]. Choice competition could occur independently in ALM and SC, or SC could dictate the state of choice competition in both regions. We next used transiently perturbations to further resolve the relationship between ALM and SC. Transient perturbations can provide a powerful tool to probe the functional organization of complex neural networks. For example, if multiple network nodes independently mediate choice competition and maintain choice activity, dynamics will quickly recover after a transient perturbation to single network nodes because other redundant nodes recover the state of choice activity based on their own activity states[55,61]. Alternatively, if a single network node can dictate choice competition across the entire network, transiently perturbing that node alone will persistently bias the state of network choice activity with little compensation by other network nodes.

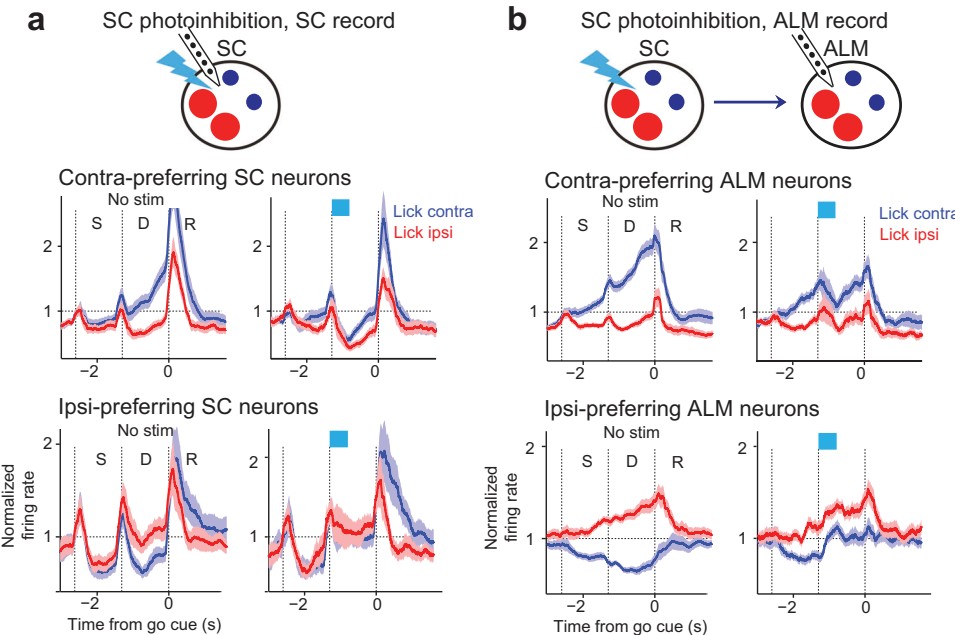

**Fig. 6 | Transient perturbation in SC persistently biases choice activity in ALM.**
**a** *Top*, schematic of SC recording during transient SC photoinhibition. *Middle*, comparison of activity in contra-preferring SC neurons during control (*left*) and photostimulation (*right*). *Bottom*, activity in ipsi-preferring SC neurons. Population response of SC neurons in lick contra trials (blue) and lick ipsi trials (red). Both correct and error trials are included, grouped by instructed lick direction relative to the recorded hemisphere. Only neurons with significant trial-type selectivity during the delay epoch are included. Neurons are sorted by their preferred trial type using spike counts from 10 trials and the remaining data is used to compute the population response. The spike rate of each neuron is normalized to the mean spike rate across all trial types. Activities of contra-preferring and ipsi-preferring neurons are first averaged within each session and the plots show mean ± s.e.m. across sessions (19 sessions, 11 mice). **b** Same as **a** but for ALM recording during transient SC photoinhibition in the same hemisphere. $N = 40$ sessions, 12 mice.

We first performed transient unilateral SC photoinhibition during the early delay epoch. We used optrode recordings to examine the effect of photoinhibition on choice activity within SC (Fig. 6a). Photoinhibition suppressed the activity of contra-preferring population while enhancing the activity of ipsi-preferring population (Supplementary Fig. 7a), thus rendering the activity to be similar to ipsilateral choice dynamics. After the photoinhibition, the activity remained in the biased state for the reminder of the delay (Fig. 6a and Supplementary Fig. 7a; non-trial-type specific change $\beta_0$, $p = 0.08$ and 0.80, trial-type specific change $\beta_1$, $p < 0.001$ and $p < 0.05$ for contra-preferring and ipsi-preferring populations respectively), consistent with the biased behavioral choice (Fig. 1h). In ALM, transient SC photoinhibition during the early delay epoch similarly biased the activity of contra- and ipsi-preferring populations. Importantly, the activity remained biased for the remainder of the delay epoch following photostimulation (Fig. 6b and Supplementary Fig. 7b; non-trial-type specific change $\beta_0$, $p = 0.16$ and 0.96, trial-type specific activity change $\beta_1$, $p < 0.001$ and $p < 0.001$). This indicates that SC provided a driving force for the competition between contra- and ipsi-preferring populations in ALM and no additional driving forces could correct the impact of SC stimulation on choice activity.

As a comparison, we also unilaterally photoinhibited ALM while recording from SC (Supplementary Fig. 7c-d). We previously found that transient unilateral ALM photoinhibition does not persistently alter ALM choice activity due to compensation from the other hemisphere[55,61]. Here we examine whether SC is also robust to unilateral ALM photoinhibition. We unilaterally photoinhibited ALM during the early delay epoch and we used the same photostimulation power as the SC photoinhibition (1.2–1.5 mW) to permit a direct comparison to SC photostimulation. This photostimulation power was sufficient to silence ALM activity through all layers[62]. ALM photoinhibition transiently reduced selectivity in SC on the same hemisphere. However, choice selectivity was only slightly reduced at the

end of the delay epoch and was not significantly different from the control trials (Supplementary Fig. 7c, d; non-trial-type specific change $\beta_0$, $p = 0.89$ and $p = 0.53$, trial-type specific change $\beta_1$, $p = 0.07$ and $p = 0.85$). Transient unilateral ALM photoinhibition also minimally affected behavioral choice (Supplementary Fig. 7c). These data show that transient unilateral ALM photoinhibition produced less persistent effects on SC choice activity and choice behavior. Unilateral loss of ALM choice activity was likely compensated by redundant choice activity in the other ALM hemisphere[55,61] and SC.

Together, these data show that ALM choice activity could not overcome a transient unilateral SC perturbation. We note that finding SC could persistently alter network choice activity does not rule out other sites within the action selection network that could also bias choice activity. Although choice activity is robust to unilateral ALM photoinhibition, strong bilateral perturbation of ALM has been shown to persistently abolish choice activity[44,55]. In addition, perturbations in the basal ganglia[7,33,63] and cerebellum[52] can also bias choice activity. These regions may form recurrent loops with SC and work together to mediate choice activity (see discussion). Notwithstanding, our results highlight SC as a key locus for driving choice activity within recurrent multi-regional networks, where modulation of SC could powerfully alter the state of network choice activity with little compensation by other network nodes.

## SC cell types bidirectionally drive choice competition
Equal proportions of SC neurons preferred contralateral or ipsilateral choice prior to the licking response (Fig. 3a-b), yet photoactivation of SC GABAergic neurons or glutamatergic neurons bidirectionally biased choice activity (Fig. 4a-c). We examined whether choice encoding might be non-uniformly distributed across SC cell types during action selection.

We analyzed SC optrode recordings during SC photoinhibition in VGAT-ChR2-EYFP mice to identify neurons excited or inhibited by

photostimulation (Supplementary Fig. 8a). We then examined their choice preference in trials without photostimulation (Supplementary Fig. 8b, c). In total, we obtained 81 neurons from the central region of SC with significant selectivity during the delay epoch. A subset of these neurons (15/81) was excited by photostimulation (Supplementary Fig. 8a), which were presumably GABAergic neurons. The excited neurons were predominantly ipsi-preferring (Supplementary Fig. 8b, $p < 0.05$, bootstrap, Methods) and exhibited robust ipsilateral selectivity during the delay epoch at the level of population-averaged activity (Supplementary Fig. 8b, spike count difference between lick contra and lick ipsi trials; $p = 0.001$, one-tailed t-test against 0). Another population of neurons were inhibited by photostimulation (38/81, Supplementary Fig. 8a), which likely included glutamatergic neurons. The inhibited neurons were predominantly contra-preferring (Supplementary Fig. 8c, $p < 0.01$, bootstrap) and showed robust contralateral selectivity during the delay epoch ($p < 0.05$, one-tailed t-test against 0). These observations suggest that SC GABAergic neurons preferentially encoded ipsilateral choice and inhibited glutamatergic neurons encoding contralateral choice.

Long-duration photostimulation of SC could induce activity changes through long-range pathways with complex temporal dynamics. To further verify SC cell-type identity and examine their choice encoding, we recorded selectively from GABAergic and glutamatergic neurons using ChR2 tagging. We first performed ChR2-tagging of SC GABAergic neurons (Fig. 7a). During optrode recordings in VGAT-ChR2-EYFP mice or GAD2-ires-cre mice expressing ChR2 in GABAergic neurons, we photostimulated SC using 1-ms light pulses (Methods). Recordings were targeted to the central region of SC where choice-selective neurons were enriched. Photostimulation reliably triggered action potentials with short latency and small temporal jitter in subsets of SC neurons (Fig. 7a and Supplementary Fig. 8d, e). These neurons were deemed to be GABAergic neurons. In total, we identified 56 GABAergic neurons out of 329 neurons in 7 mice. In the delayed response task, the identified GABAergic neurons predominantly showed ipsilateral choice preference during the delay epoch (Fig. 7b and Supplementary Fig. 8f; fraction of contra- vs. ipsi-preferring neurons, $p < 0.01$, bootstrap). As a population, the GABAergic neurons exhibited a robust buildup of ipsilateral selectivity during the delay epoch (Fig. 7b; $p < 0.05$, one-tailed t-test against 0). Interestingly, SC GABAergic neurons encoded ipsilateral choice during the delay epoch, but their preferences switched to encode contralateral choice during the response epoch (Supplementary Fig. 9a-d). Thus SC GABAergic neurons may play distinct roles during action selection vs. movement initiation.

We next recorded selectively from SC glutamatergic neurons. We performed optrode recordings in Vglut2-ires-cre x Ai32 mice and used the same procedures to identify neurons reliably activated by 1-ms photostimulation. In total, we identified 60 glutamatergic neurons in 2 mice. In the delayed response task, the identified glutamatergic neurons predominantly showed contralateral choice preference during the delay epoch (Fig. 7c and Supplementary Fig. 8g; fraction of contra- vs. ipsi-preferring neurons, $p < 0.001$, bootstrap), opposite to SC GABAergic neurons. As a population, the glutamatergic neurons exhibited a buildup of contralateral selectivity during the delay epoch (Fig. 7c; $p < 0.001$, one-tailed t-test against 0).

These cell-type specific recordings show that SC GABAergic and glutamatergic neurons exhibited opposing choice selectivity prior to the licking response, with the strongest selectivity observed during the delay epoch (Fig. 7b, c). This pattern and time course of choice selectivity in SC cell types mirrored the effect of our SC manipulations: ChR2 photostimulation of GABAergic neurons biased upcoming choice to ipsilateral direction while photoactivation of glutamatergic neurons induced a contralateral bias, with the strongest bias induced during the delay epoch (Fig. 1h, i). However, ChR2 may induce supra-physiological activation of these cell types. For example, ChR2

activation of SC GABAergic neurons may simply inhibit glutamatergic neurons, which does not necessarily show the involvement of GABAergic neurons in action selection. To further test the necessity of SC cell types, we directly silenced each cell type using inhibitory opsins. We directly inhibited SC GABAergic neurons using cre-dependent ArchT viruses in GAD2-ires-Cre mice or Vgat-ires-cre mice (Supplementary Fig. 9e). Silencing SC GABAergic neurons during the delay epoch biased upcoming choice to the contralateral direction (Fig. 7d and Supplementary Fig. 9f), opposite to the bias induced by GABAergic neuron activation (Fig. 1h). We directly inhibited SC glutamatergic neurons using GtACR1 (in Vglut2-ires-cre mice crossed to a cre-dependent soma-targeted GtACR1 reporter mouse, Supplementary Fig. 9e). Silencing SC glutamatergic neurons biased upcoming choice to the ipsilateral direction (Fig. 7e), opposite to the effect of glutamatergic neuron photoactivation (Fig. 1i). Altogether, our data show that inhibiting or activating each SC cell type could bidirectional bias upcoming choice.

These results revealed a circuit mechanism for choice competition prior to the licking response: SC GABAergic neurons encode ipsilateral choice and locally inhibit glutamatergic neurons encoding contralateral choice in the same hemisphere (Fig. 7f), while disinhibiting contra-preferring neurons in the other hemisphere (Fig. 5). These SC circuits in turn drive competitive interactions between contra-preferring and ipsi-preferring populations in ALM.

## Discussion

Our anatomical and electrophysiology mappings identify a topographically organized frontal cortical and SC circuit responsible for the selection and initiation of directional licking in mice (Figs. 1–2). Within topographically matched regions of ALM and SC, two intermingled populations of neurons represent contralateral and ipsilateral choices and exhibit choice competition dynamics prior to the lick response (Fig. 3). Our data show that SC is a driver of choice activity in ALM during action selection (Figs. 4–6), and modulating SC GABAergic and glutamatergic neurons can bidirectionally drive push-pull between ALM populations coding opposing choice. These results thus highlight cell types within SC as a key network node to modulate choice competition within the broader action selection network.

During decision-making, neural activity correlated with future choice has been observed across the frontal cortex and SC[3–5,8–16,35], and earlier works show that modulating these regions on their own can bias choice at the level of behavior[12,14,15,32–37,40,42–44]. However, the relationship between brain regions and how they interact to mediate choice activity remains unresolved. Traditional theories ascribe action selection processes to the frontal cortex and basal ganglia while implicating SC to be downstream to these regions[19–22]. Other works suggest frontal cortex and SC work cooperatively[15,16,42]. Our findings challenge the notion that decision is formed exclusively in the frontal cortex and passed to SC. A transient perturbation of SC biases ALM choice activity, and in absence of further SC stimulation, ALM choice activity remains in the biased state (Fig. 6b). This argues against ALM circuit forming its own independent decision as it could not overcome the impact of transient SC perturbation.

Where might the choice activity originate? Our recordings in ALM and SC show that while activity correlated with sensory stimulus arises early during the sample epoch, activity correlated with future choice emerges late during the sample epoch (Supplementary Fig. 3). Choice activity emerges simultaneously in ALM and SC and gradually increases throughout the delay epoch with similar time course (Supplementary Fig. 3f, j). These findings mirror previous recordings comparing the latency of choice activity across the frontal cortex and SC[16]. The time course of choice activity is also consistent with the effect size of our SC photoinhibition, which produces the strongest effect on upcoming choice during the delay epoch (Fig. 1h). While our data cannot exclude another brain region carrying choice information

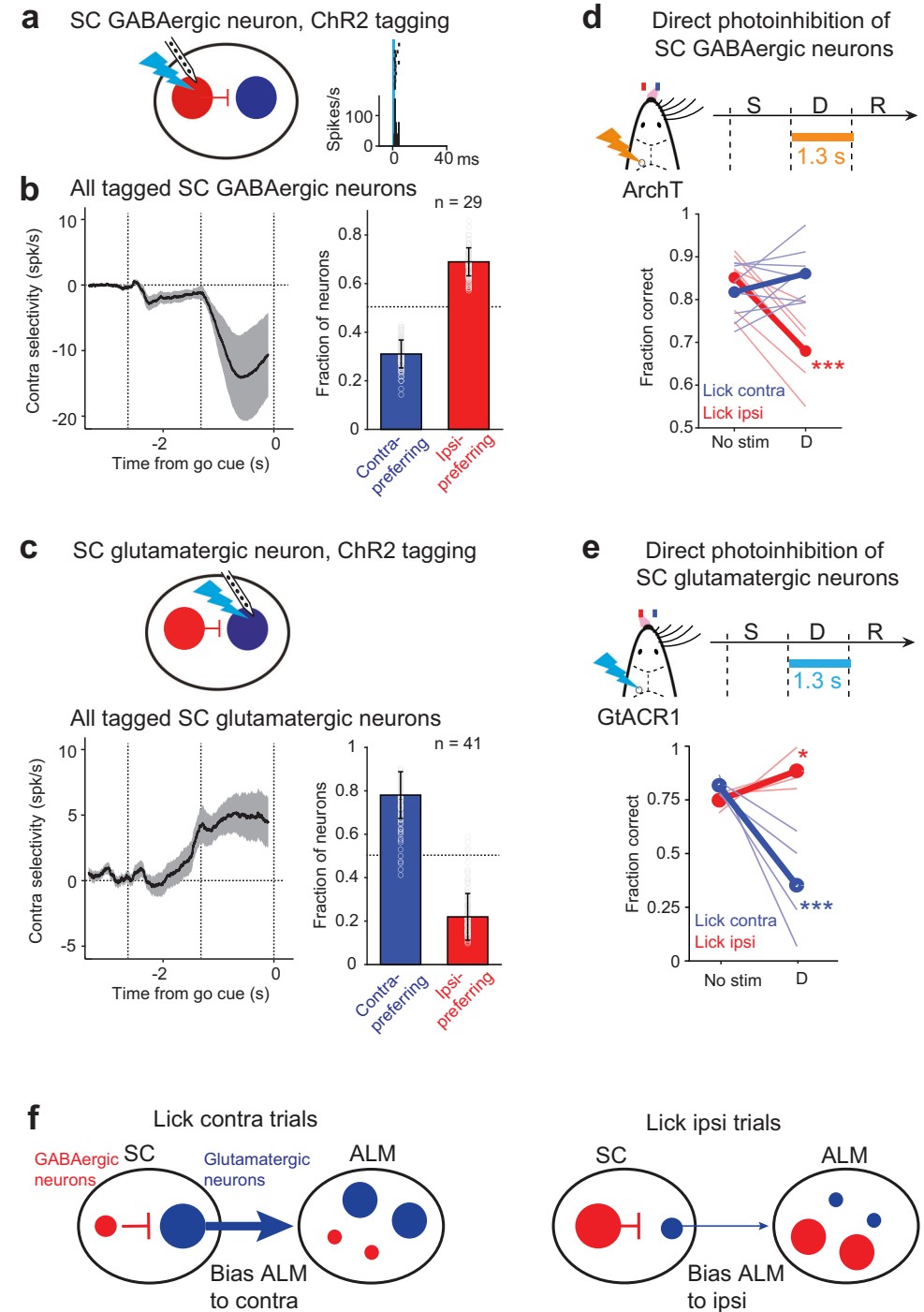

**Fig. 7 | SC cell types bidirectionally drive choice competition. a** ChR2 tagging of SC GABAergic neurons. An example tagged GABAergic neuron is shown. Raster and PSTH are aligned to photostimulus onset. Photostimulation was performed outside of the behavioral task. **b** Tagged SC GABAergic neurons are predominantly ipsi-preferring. *Left*, contra-selectivity across the SC GABAergic neurons (mean ± s.e.m.). Only neurons with significant trial-type selectivity during the delay epoch are included (*n* = 29 from 8 mice). Contra-selectivity is the spike rate difference between lick contra and lick ipsi trials. *Right*, proportions of GABAergic neurons that are contra-preferring (blue) and ipsi-preferring (red). *P* < 0.01, significantly more ipsi-preferring neurons than contra-preferring neurons, one-tailed t-test, bootstrap. Dots, fractions from bootstrap. Bar graphs represent mean ± s.e.m. **c** Tagged SC glutamatergic neurons are predominantly contra-preferring. Same as **b** but for SC glutamatergic neurons (*n* = 41 from 2 mice). *P* < 0.001, significantly more contra-preferring neurons than ipsi-preferring neurons, one-tailed t-test,

bootstrap. Bar graphs represent mean ± s.e.m. **d** Direct silencing of SC GABAergic neurons during the delay epoch biases future choice to the contralateral direction. AAV-flexed-ArchT viruses in SC of 6 GAD2-ires-cre mice and 2 Vgat-ires-cre mice. Thick lines, mean; thin lines, individual mice (*n* = 8). **p < 0.001; one-tailed t-test, bootstrap (Methods). Trials are grouped by instructed lick direction relative to the manipulated hemisphere. Blue, contralateral (lick contra); red, ipsilateral (lick ipsi). Photostimulation power, 10 mW. **e**. Direct silencing of SC glutamatergic neurons biases future choice to ipsilateral direction. Vglut2-ires-cre x GtACR1 mice. *N* = 4 mice. Photostimulation power, 0.05–0.2 mW. **f**. A model of ALM-SC circuit for action selection. SC GABAergic neurons promote ipsilateral choice and glutamatergic neurons promote contralateral choice, and the opposing actions of these SC cell types drive push-pull between contra-preferring and ipsi-preferring populations in ALM.

earlier than SC and ALM, these data are consistent with mice forming a decision starting from the late sample epoch, and SC drives the gradual buildup of choice activity. The addition of a delay epoch with a fixed duration may have caused the slow emergence of choice activity in the delayed response task. These data do not yet clarify what drives the gradual buildup of choice activity in SC and whether the decision-formation process involves potential upstream regions.

SC forms a re-entrant loop with ALM via the thalamus (Fig. 1c, d)[53,54]. ALM-projecting thalamus also receives inputs from the basal ganglia and cerebellum[51,52]. These regions may form recurrent loops with ALM and SC to collectively mediate choice activity. Manipulations of the striatum and substantia nigra pars reticulata (SNr) can opposingly shape push-pull choice activity and bias future lick direction[7,33,63,64]. Manipulation of the cerebellum alters ALM choice activity and causes a global reorganization of ALM population activity[52]. Finally, our previous analysis found that a complete bilateral silencing of ALM activity can also abolish subsequent choice activity, resulting in random directional biases[44,55]. It is worth noting that perturbing these network nodes also affects the activity of SC via the cortico-collicular projections[14,41] or SNr-SC projections[25,33]. For example, bilateral inactivation of ALM may very well disrupt behavior as well as delay period activity in SC, consistent with our previous work[55], and is an open question. Within this recurrent multiregional network, our results highlight SC as a major locus that can bidirectionally and persistently bias the state of network choice activity, with little compensation by other network nodes (Fig. 6a, b). It remains to be determined if there are network nodes that could modulate choice competition independent of SC. We propose that SC could provide a mechanism for network activity to influence choice competition. For example, inputs from other brain regions could influence choice competition by modulating SC activity. SC output in turn drives the state of network choice activity. This looped network architecture could make the current state of decision continuously available to other network nodes for specific computations.

SC GABAergic neurons locally inhibit glutamatergic neurons[65,66]. Our findings suggest that these SC cell types form a functional circuit during action selection whereby SC GABAergic neurons encode ipsilateral choice and inhibit glutamatergic neurons which encode contralateral choice. The glutamatergic neurons further project to ALM via thalamus (Fig. 7f). Modulating these SC cell types can bidirectionally drive choice competition activity across the action selection network. The opposing actions of SC cell types likely reflect their local inhibitory connectivity. Intriguingly, the opposing choice encoding of SC cell types is only observed during the delay epoch whereas during the response epoch both cell types exhibit congruent preference for contralateral choice (Supplementary Fig. 9). Manipulating the SC GABAergic neurons during the response epoch also does not produce bidirectional effects on choice, in contrast to the delay epoch (compare Supplementary Figs. 2d and 9f). More work is needed to understand how choice-related activity impinges upon SC cell-types and how these neural populations interact across action selection and motor response phases of the behavior.

SC likely modulates choice activity in ALM via the thalamus[51,56]. Intriguingly, SC projections to the thalamus are primarily glutamatergic[14]. Yet, photoinhibiting SC induced coordinated excitation and inhibition across contra-preferring and ipsi-preferring neurons in ALM to drive their push-pull (Fig. 4). SC may provide excitation selectively to the contra-preferring neurons in thalamo-cortical loop and inhibit the ipsi-preferring neurons through disynaptic inhibition. Possible candidates for this inhibition include cortical interneurons and thalamic reticular nucleus. How SC inputs interact with circuit dynamics within thalamo-cortical loop to produce push-pull modulation remains to be determined.

In addition, competition between choice options likely occurs across hemispheres. Activating SC GABAergic neurons locally inhibit contra-preferring neurons within the same hemisphere (Fig. 7f) and disinhibits contra-preferring neurons in the other hemisphere (Fig. 5), which suggests interhemispheric mutual inhibition that coordinates choice competition in both hemispheres[33,60]. Interhemispheric inhibition could occur either at the level of SC or ALM. We previously found that perturbing each hemisphere of ALM does not strongly influence the choice activity of the other ALM hemisphere[61], which suggests weak coupling between ALM hemispheres. Consistent with this notion, choice activity and behavioral performance are robust to transient unilateral ALM photoinhibition (Supplementary Fig. 7c-d). Thus, mutual inhibition that coordinates network choice activity likely occurs directly between SC hemispheres. SC interhemispheric coordination could be mediated by SC commissural inhibitory projections[14,60,65,66], SC excitatory projections to inhibitory neurons in the other hemisphere[14,65], or inhibitory pathways outside of SC[67,68].

Previous studies found lateralized representation of contralateral motor choice in SC[16,33,60], including in SC GABAergic neurons[60]. However, these analyses examine SC activity in tasks without a delay epoch[33,60] or in time windows immediately prior to the motor response[16]. In contrast, other studies examining SC selectivity in tasks with a delay epoch found equal proportions of contra- and ipsi-preferring neurons prior to the motor response[14,15]. We found equal proportions of SC neurons encode contralateral and ipsilateral choices during the delay epoch, but immediately after the go cue SC neurons become overwhelmingly contra-preferring (Fig. 3b and Supplementary Fig. 9a, b). Cell-type specific recordings from SC GABAergic neurons show that these neurons encode ipsilateral choice during the delay epoch but switch their preference to encode contralateral choice during the response epoch (Supplementary Fig. 9c, d). Similar switching selectivity in SC GABAergic neurons has been reported during auditory decision-making in mice[14]. These data suggest that SC GABAergic neurons play different roles during action selection and movement initiation (Supplementary Fig. 9f). It also highlights a need to disentangle activity related to action selection from motor response.

Our study examines SC during action selection of directional licking. It remains to be tested whether SC supports action selection beyond lateralized licking movements. Recent studies have identified that SC encodes three-dimensional head movements[69]. Furthermore, previous inactivation of SC found impaired action selection toward spatial targets across different motor modalities, including eye movement and arm movement in primates[36,37], orienting and licking in rodents[12,14,15,42], as well as impairing other forms of selection such as spatial visual attention[38–40,70–72]. Stimulation of SC in rodents can drive diverse motor responses, including orienting, freezing, locomotion, jumping[73–78], and licking (Fig. 1). SC may provide a general circuit motif for selection of competing potential actions.

## Methods

### Mice

This study was based on data from 104 mice (age > postnatal day 60, both male and female mice). Fourteen wildtype and EMX1-ires-cre mice (JAX 005628) were used for anatomical tracing. Three wildtype mice with pAAV-hSyn-hChR2(H134R)-EYFP injected in the lateral superior colliculus (SC) and 2 Vglut2-ires-cre mice with a cre-dependent ChR2 virus injected in lateral SC were tested for SC photostimulation to evoke licking. Ten Vglut2-ires-cre mice (JAX 016963) crossed with Ai32 mice (JAX 012569, Vglut2-ires-cre × Ai32) were used for SC photo-activation experiment to test SC involvement in action selection. A subset of these mice (4 mice) was used for ALM recordings during SC photoactivation. Another subset of the Vglut2-ires-cre × Ai32 mice (2 mice) was used for SC optrode recordings and ChR2 tagging of glutamatergic neurons. 36 VGAT-ChR2-EYFP mice (JAX 014548, derived using a VGAT-mhChR2-YFP BAC transgene[79]) were used for ALM or SC photoinhibition[14]. A subset of these mice (13 mice) was used for SC

optrode recordings and ChR2 tagging of GABAergic neurons. Another subset of these mice (12 mice, including 3 mice used for SC optrode recordings) were used for ALM recordings during SC photoinhibition. Another subset of these mice (6 mice) was used for SC recordings during ALM photoinhibition. Five GAD2-ires-cre mice (JAX 010802) were used for SC photoinhibition experiment using cre-dependent ChR2 virus injection. One of these mice was also used for SC optrode recordings and ChR2 tagging of GABAergic neurons. Six additional GAD2-ires-cre mice and two Vgat-ires-cre mice were used for cre-dependent ArchT virus injection and photoinhibition of SC GABAergic neurons. Four Vglut2-ires-cre mice crossed with a cre-dependent GtACR1 mice (JAX 033089, R26-LNL-GtACR1-Fred-Kv2.1) were used for photoinhibition of SC glutamatergic neurons. A subset of these mice (3 mice) was used for recordings in the opposite SC hemispheres during photoinhibition. Twenty-two additional mice (5 PV-Cre × Rosa26-LSL-ReaChR, JAX 008069 and 024846, 14 VGAT-ChR2-YFP, 3 PV-Cre × Ai32), were used for ALM and SC recordings during behavior.

All procedures were in accordance with protocols approved by the Institutional Animal Care and Use Committees at Baylor College of Medicine. Mice were housed at a constant temperature (22 ± 1 °C) and humidity (30-55%) under a 12:12 reverse light:dark cycle and tested during the dark phase. On days not tested, mice received 0.5–1 mL of water. On other days, mice were tested in experimental sessions lasting 1 to 2 hours where they received all their water (0.3 to 1 mL). If mice did not maintain a stable body weight, they received supplementary water[80]. All surgical procedures were carried out aseptically under 1-2% isoflurane anesthesia. Buprenorphine Sustained Release (1 mg/kg) and Meloxicam Sustained Release (4 mg/kg) were used for pre- and post-operative analgesia. A mixture of Bupivacaine and Lidocaine was administered topically before scalp removal. After surgery, mice were allowed to recover for at least three days with free access to water before water restriction.

## Surgery
Mice were prepared for photostimulation and electrophysiology with a clear-skull cap and a headpost[43,80]. The scalp and periosteum over the dorsal surface of the skull were removed. A layer of cyanoacrylate adhesive (Krazy glue, Elmer's Products Inc) was directly applied to the intact skull. A custom-made headpost was placed on the skull (approximately over visual cortex) and cemented in place with clear dental acrylic (Lang Dental Jet Repair Acrylic; Part# 1223-clear). A thin layer of clear dental acrylic was applied over the cyanoacrylate adhesive covering the entire exposed skull, followed by a thin layer of clear nail polish (Electron Microscopy Sciences, Part# 72180).

In mice prepared for SC photostimulation experiments, a 2 mm long optical fiber (Thorlabs, Part#: CFML12L02) was implanted to target the lateral SC in either the left or right hemisphere (posterior 3.5 mm from bregma, lateral 1.5 mm). Optical fibers were implanted either unilaterally to target single SC hemispheres or bilaterally to target both SC hemispheres. In GAD2-ires-cre mice prepared for SC photoinhibition experiment, 300 nL of AAV1-CAGGS-Flex-ChR2-tdTomato virus (Penn Vector Core, 1.38×10^13 vg/mL) was injected in the lateral SC (posterior 3.5 mm from bregma, lateral 1.5 mm, depth 2.5 mm), followed by implantation of an optical fiber over the injection site. In GAD2-ires-cre mice or Vgat-ires-cre mice prepared for photoinhibition of SC GABAergic neurons, 200-500 nL of AAV5-Flex-ArchT-tdTomato (Addgene, 1.6×10^13 vg /mL) was injected in the lateral SC at the same coordinates followed by implantation of an optical fiber over the injection site.

## Viral injection and histology
Injection pipettes were pulled from glass capillary micropipettes (Wiretrol II, Drummond Scientific Company) using P-97 (Sutter Instrument Company). The tip was 20-30 μm in diameter and beveled. Pipettes were back-filled with mineral oil and front-loaded with viral suspension immediately before injection. Injections were made through the thinned skull using a custom, piston-based, volumetric injection system.

To characterize ALM descending projections, AAV viruses carrying fluorescent proteins were injected in the medial and lateral regions of left ALM (medial ALM: anterior 2.5 mm from bregma, lateral 1 mm; lateral ALM: anterior 2.5 mm, lateral 2 mm; 60-150 nl at depth 0.75 mm). AAV viruses were pENN.AAV.CAG.tdTomato.WPRE.SV40 (Addgene, 105554-AAV1, 1.9×10^13 vg/mL), AAV9-syn-RFP (SignaGen, SL116027, 1.68×10^13 vg/mL), AAV-pCAG-FLEX-EGFP-WPRE (Addgene, 51502-AAV1, 1.9×10^13 vg/mL), and pAAV-hSyn-EGFP (Addgene, 50465-AAV1, 1.1×10^13 vg/mL). In different brains, red (tdTomato or RFP) or green (GFP) fluorescent proteins were used for either the medial ALM or lateral ALM in a counter-balanced manner. Incubation period was 11-21 days before perfusion.

To characterize SC descending projections in IRt and ascending projections in the thalamus, 120-300 nL of pENN.AAV.CAG.tdTomato.WPRE.SV40 or AAV9-syn-RFP was injected in the lateral region of left SC (posterior 3.5 mm from bregma, lateral 1.5 mm, 120-300 nl at depth 2.5 mm). The incubation period was 10-21 days before perfusion. To map connectivity between the lateral SC and ALM in the thalamus, the ALM-projecting thalamus was labeled in the same brain. Retrograde tracer wheat germ agglutinin (WGA) (Thermo Fisher Scientific, WGA-Alexa488, 2% in PBS) was injected in the left ALM (300 nl at depth 0.75 mm). WGA injection was performed 24 hours before perfusion.

To label SC IRt-projecting neurons and thalamus-projecting neurons, retrograde tracers were injected either in the left thalamus targeting the ventral medial nucleus (VM, posterior 1.3 mm from bregma, lateral 0.8 mm, 300 nl at depth 4.3 mm) or in the right IRt (posterior −6.2 mm from bregma, lateral 1.0 mm, 300 nl at depth 4.2 mm). Retrograde tracers were cholera toxin subunit B (CTB-488; Alexa 488; Molecular probe, Invitrogen, 0.5% in HEPES buffered saline), WGA-Alexa488 (Thermo Fisher Scientific, 2% in PBS), WGA-Alexa594 (Thermo Fisher Scientific, 2% in PBS), and pAAV-CAG-GFP (Addgene, 37825-AAVrg, 7.0×10^12 vg/mL). In a subset of the brains, VM and IRt injections were made in the same brain. WGA-Alexa594 and pAAV-CAG-GFP were used for either VM or IRt in a counter-balanced manner across different brains. The incubation time was 24 hours for WGA and 14 days for pAAV-CAG-GFP before perfusion.

Mice were perfused transcardially with PBS followed by 4% paraformaldehyde (PFA)/ 0.1 M PBS. The brains were fixed overnight and transferred to 20% sucrose before sectioning on a freezing microtome. Coronal 50 μm free-floating sections were processed using standard fluorescent immunohistochemical techniques. Slide-mounted sections were imaged with a Zeiss microscope, a 10× objective and a Hamamatsu Orca Flash 4 camera. Each coronal section was made up of 80–200 tiles merged with Neurolucida software[81].

For ArchT-mediated direct photoinhibition of SC GABAergic neurons, we performed immunohistochemistry to verify that the ArchT expression was specific to GABAergic neurons (Supplementary Fig. 9e). For GtACR1-mediated direct photoinhibition of SC glutamatergic neurons, we performed immunohistochemistry to verify that the GtACR1 expression did not include GABAergic neurons (Supplementary Fig. 9e). Mice were perfused as above but the brains were only fixed for 45 minutes before transferring to 30% sucrose. Coronal 40 μm free-floating sections were collected with a cryostat. Slices were incubated with standard 10% goat serum (Gibco sera) for half an hour before incubation in the primary anti-GABA antibody (rabbit, dilution 1:1000, A2052; Sigma) for 48 hours. After washing 3 times (10 minutes each time) with PBS, sections were incubated in the secondary antibody (donkey anti-rabbit, Alexa Fluor 488, dilution 1:1000, A-21206; ThermoFisher Scientific) for 2 hours. Slices were mounted with DAPI medium and imaged with a LSM780 confocal microscope (x20 objective).

## Behavior

Behavioral data was acquired using Bpod (https://www.sanworks.io) and wavesurfer (v 0.787) (https://wavesurfer.janelia.org/). The behavioral task and training have been described[80,82]. The stimulus was a metal pin (0.9 mm in diameter), presented at one of two possible positions (Fig. 1e). The two pole positions were 5 mm apart along the anterior-posterior axis and were constant across sessions. The pole was positioned 5 mm lateral from the whisker pad. The posterior pole position targeted the C2 whisker when whiskers were at their resting positions. The pole made contacts with multiple whiskers at both positions, typically with a different set of whiskers. A two-spout lickport (4.5 mm between spouts) was used to deliver water rewards and record licks.

At the beginning of each trial, the vertical pole moved into reach of the whiskers (0.2 s travel time), where it remained for 1 second, after which it was retracted (retraction time 0.2 s). The sample epoch was defined as the time between the pole movement onset to 0.1 s after the pole retraction onset (sample epoch, 1.3 s, Fig. 1e). The delay epoch (durations, 1.3 s) followed the sample epoch. An auditory 'go' cue indicated the end of the delay epoch (pure tone, 3.4 kHz, 0.1 s duration). Licking early during the trial was punished by a loud alarm sound (siren buzzer, 0.05 s duration), followed by a brief timeout (1-1.2 s). Licking the correct lickport after the 'go' cue led to a water reward (2-3 μL). Licking the incorrect lickport triggered a timeout (2-6 s). Trials in which mice did not lick within a 1.5 second window after the 'go' cue ('ignore') were rare and typically occurred at the end of a session. Reaction time was defined from the 'go' cue onset to the first lickport contact.

## Videography

Two CMOS cameras (CM3-U3-13Y3M, FLIR) were used to measure orofacial movements of the mouse under IR illumination (940 nm, Roithner Laser, LED940-66-60). One camera acquired the bottom view of the mouse with a 4–12 mm focal length lens (12VM412ASIR, Tamron) and pixel resolution of 0.065 mm/pixel. The second camera acquired the side view of the mouse with a 4–12 mm focal length lens (12VM412ASIR, Tamron) and pixel resolution of 0.07 mm/pixel. Videos were acquired at 200 Hz framerate using FlyCapture (FLIR).

## Photostimulation

**SC photoactivation.** For SC photoactivation to test the SC involvement in the delayed response task, we photostimulated SC glutamatergic neurons in Vglut2-ires-cre × Ai32 mice or in VGlut2-ires-cre mice with AAV1-CAGGS-Flex-ChR2-tdTomato virus injected in the lateral SC (posterior 3.5 mm from bregma, lateral 1.5 mm, 300 nl at depth 2.5 mm). SC glutamatergic neurons, but not GABAergic neurons, provide outputs to the thalamus and medulla[14]. Light was delivered to the lateral SC through an optical fiber. Either the left or right SC hemisphere was tested in different mice. Light from a 473 nm laser (UltraLasers, MBL-FN-473-300mW) was controlled by an acousto-optical modulator (AOM; Quanta Tech, MTS110-A3-VIS) and a shutter (Vincent Associates). To prevent the mice from distinguishing photostimulation trials from control trials using visual cues, a 'masking flash' was delivered using 470 nm LEDs (Luxeon Star) near the eyes of the mice. The masking flash began as the pole started to move and continued through the end of the epoch in which photostimulation could occur. Photostimulation was deployed on 25% of the behavioral trials. In some sessions, we recorded from ALM using silicon probes during SC photoactivation. In those sessions, photostimulation was deployed on 20-40% of the trials.

We used 40 Hz photostimulation with a sinusoidal temporal profile. The duration was 500 ms or 1.3 s, including a linear ramp during laser offset (100 ms). The average power at the fiber tip was 0.1–0.2 mW. High power SC photoactivation could induce contralateral licking. We therefore chose a laser power for each mouse in which the early lick rate was low. 3 mice were tested with 0.2 mW and 1 mouse was tested with 0.1 mW. We did not observe difference between these mice. We performed photostimulation during either the sample, delay, or response epochs (1.3 s; Supplementary Fig. 2g-h), as well as during sub-epochs of the sample and delay epochs (the first or last 500 ms of each epoch, Fig. 1i). Whole-epoch photoactivation was designed to probe the contribution of SC to action selection and movement initiation. Sub-epoch manipulation was designed to further probe the contribution of SC to action selection specifically. If SC is involved in action selection, transient SC photoactivation should bias future choice even after the cessation of the photostimulus.

In a separate group of mice, a wider range of power was used to evoke contralateral licking during the delayed response task (0.1-0.8 mW; Fig. 1f and Supplementary Fig. 2b). These experiments examined the role of SC in eliciting licking movement in general. These experiments were performed in Vglut2-ires-cre mice with AAV1-CAGGS-Flex-ChR2-tdTomato virus injected in the lateral SC or in wildtype mice with pAAV-hSyn-hChR2(H134R)-EYFP virus injected in the lateral SC (posterior 3.5 mm from bregma, lateral 1.5 mm, 300 nl at depth 2.5 mm). Either the left or right SC hemisphere was tested in different behavioral sessions. In a subset of these mice, SC photoactivation was also tested outside of the delayed response task (0.5 mW–8 mW; Supplementary Fig. 2a). At the time of testing, the mice were already trained in the behavioral task, but the mice were tested in absence of sensory stimulus and reward. The mice were also nonwater restricted.

**SC photoinhibition.** For SC photoinhibition, we photostimulated SC GABAergic neurons in VGAT-ChR2-EYFP mice or in GAD2-ires-cre mice with AAV1-CAGGS-Flex-ChR2-tdTomato virus injected in the lateral SC (posterior 3.5 mm from bregma, lateral 1.5 mm, 300 nl at depth 2.5 mm). SC GABAergic neurons locally inhibit other SC neurons, including glutamatergic neurons that project to the thalamus and medulla. Subpopulations of SC GABAergic neurons also send long-range inhibitory projections to the other SC hemisphere[28,66], or outside of SC[66], but they do not project to the thalamus and brainstem[14]. 473 nm light was delivered to the lateral SC through an optical fiber. Either the left or right SC hemisphere was tested in different mice. In a subset of mice, the optical fiber was implanted in both SC hemispheres to bilaterally inhibit SC. Photostimulation was deployed on 20-40% of the behavioral trials. During ALM recordings, photostimulation was deployed on 40% of the trials.

We used 40 Hz photostimulation with a sinusoidal temporal profile. For unilateral SC photoinhibition, the photostimulus duration was either 500 ms (sub-epoch photoinhibition) or 1.3 s (whole-epoch photoinhibition) including a linear ramp during laser offset (100 ms). Similar to the rationale for SC photoactivation, SC sub-epoch photoinhibition was during the first or last 500 ms of the sample or delay epoch (Fig. 1h); SC whole-epoch photoinhibition was during the sample, delay, or response epoch (Supplementary Fig. 2d-f). The average power at the fiber tip was 1.2 mW. For bilateral SC photoinhibition, the photostimulus duration was 1.3 s and occurred during either the sample, delay, or response epoch. The average power was 2.25 mW in each hemisphere.

**Direct photoinhibition of SC GABAergic neurons.** To directly photoinhibit SC GABAergic neurons, we used GAD2-ires-cre mice or Vgat-ires-cre mice with AAV5-Flex-ArchT-tdTomato virus injected in the lateral SC (posterior 3.5 mm from bregma, lateral 1.5 mm, 200 or 500 nl at depth 2.5 mm). Immunohistochemical staining confirmed that ArchT expression was specific to GABAergic neurons within SC (Supplementary Fig. 9e). Light was delivered to the lateral SC through an optical fiber. Either the left or right SC hemisphere was tested in different mice. 593.5 nm light from a laser (UltraLasers, MGL-N-593.5-200mW) was controlled by an acousto-optical modulator (AOM;

Quanta Tech, MTS110-A3-VIS) and a shutter (Vincent Associates). Photostimulation was deployed on 25% of the behavioral trials.

We used 40 Hz photostimulation with a sinusoidal temporal profile. The photostimulus duration was 1.3 s, including a linear ramp during laser offset (100 ms). Photostimulation was during the sample, delay, or response epoch (Supplementary Fig. 9f). The average power at the fiber tip was 10 mW.

**Direct photoinhibition of SC glutamatergic neurons.** To directly photoinhibit SC glutamatergic neurons, we used Vglut2-ires-cre × GtACR1 transgenic mice expressing a soma-targeted GtACR1 in SC glutamatergic neurons. Immunohistochemical staining confirmed that GtACR1 expression did not include GABAergic neurons within SC (Supplementary Fig. 9e). 473 nm light was delivered to the lateral SC through an optical fiber. Either the left or right SC hemisphere was tested in different mice. Photostimulation was deployed on 25% of the behavioral trials.

We used 40 Hz photostimulation with a sinusoidal temporal profile. The photostimulus duration was 1.3 s including a linear ramp during laser offset (100 ms). Photostimulation was during the delay epoch (Fig. 7e). The average power at the fiber tip was 0.05-0.2 mW.

**ALM photoinhibition.** For photoinhibition of ALM, we photo-stimulated cortical GABAergic neurons in VGAT-ChR2-EYFP mice. Photostimulation was performed by directing a 473 nm laser over the surface of the brain through a clear skull implant (beam diameter: 400 μm at 4σ). Photostimulation was directed to either the left or right ALM hemisphere and was always the same as the recorded SC hemisphere. We used the same photostimulus as for the transient SC photoinhibition to enable a direct comparison, i.e., 40 Hz photo-stimulation with a sinusoidal temporal profile, 500 ms duration (sub-epoch photoinhibition) including a linear ramp during laser offset (100 ms). Photoinhibition was during the first 500 ms of the delay epoch. The average power at the brain surface was 1.2-1.5 mW. At this photostimulation power, a single laser spot silenced 90% of spikes in a cortical area of 1 mm radius (at half-max) through all cortical layers[62]. During electrophysiology, photoinhibition was deployed on 40% of the trials to obtain a large number of trials per condition.

**Electrophysiology.** Extracellular spikes were recorded using 64-channel Cambridge NeuroTech silicon probes (H2 acute probe, 32 sites per shank spaced at 25 μm, 2 shanks spaced at 350 μm). The 64-channel voltage signals were amplified and digitized on an Intan RHD2164 64-Channel Amplifier Board (Intan Technology) at 16 bit, recorded on an Intan RHD2000-Series Amplifier Evaluation System (sampling at 20,000 Hz), and stored for offline analysis.

In each mouse, we recorded from both ALM and SC, but each region was sampled in different sessions. A small craniotomy (diameter, <1 mm) was made one day prior to the recording session. A silicon probe was acutely inserted prior to the start of the recording session. To minimize brain movement, a drop of silicone gel (3-4680, Dow Corning, Midland, MI) was applied over the craniotomy after the electrode was in the tissue. The tissue was allowed to settle for several minutes before the recording started. One craniotomy targeted a single brain region, and 4-9 recordings (also referred to as penetrations) were made from each craniotomy across different daily sessions (1 recording per day). A new craniotomy was opened only after the previous craniotomy had been sampled. Across multiple craniotomies, recordings were made from both hemispheres.

For ALM recordings, the craniotomy was centered at 2.5 mm anterior and 1.5 mm lateral from bregma. Silicon probe was inserted 0.95-1.5 mm below the brain surface, and the 2 shanks were oriented along the medial-lateral axis. For SC recordings, the craniotomy was centered at 3.5 mm posterior and 1.5 mm lateral from bregma. Silicon probe was inserted 2.25-3.0 mm below the brain surface, and the

2 shanks were oriented along the anterior-posterior axis. In most recording sessions, Dil, DiR, or DiO was applied to the tip of the silicon probe to label the recording tracks (Figs. 2-3). Recording locations were reconstructed *post-hoc* based on the florescent labeling and the activity lamination pattern across the electrodes to register the recorded units in the Allen Mouse Common Coordinate Framework (CCF)[83].

To examine the effect of SC photoinhibition on SC activity (Figs. 4-6), we performed SC recordings in VGAT-ChR2-EYFP mice using optrodes (Cambridge Neurotech, ASSY-77 H2 with Lambda-b Fiber). Recording and photostimulation procedures were the same as above. Photostimulation started at the start of the delay epoch and lasted for 0.5 s or 1.3 s. The average power at the fiber tip was 1.2 mW. In some of these recordings, we also performed ChR2 tagging of GABAergic neurons. In addition, ChR2 tagging was carried out using optrodes in additional VGAT-ChR2-EYFP mice and a GAD2-ires-cre mouse injected with AAV1-CAGGS-Flex-ChR2-tdTomato virus. To identify SC GABAergic neurons, 3 laser pulses (1 ms duration, 2.5-5.0 mW peak power at the fiber tip, separated by 200 ms) were deployed during inter-trial intervals to elicit responses from ChR2+ neurons. Photostimulation was deployed during 50% of the intertrial intervals. Photostimulation occurred well after the completion of the licking response in the previous trial (3 s after the response epoch) and well before the start of the next trials (>3 s).

**Behavioral data analysis.** We separately computed the task performance for "lick right" and "lick left" trials. Performance was computed as the fraction of correct choices, excluding lick early trials and no lick trials. Significance of the performance change in each photostimulation condition was determined using a nested bootstrap to account for variability across mice, sessions, and trials[43]. We tested against the null hypothesis that the performance change caused by photostimulation was due to normal behavioral variability. In each round of bootstrap, we replaced the original behavioral dataset with a re-sampled dataset in which we re-sampled with replacement from: 1) mice, 2) sessions performed by each mouse, 3) the trials within each session. We then computed the performance change on the re-sampled dataset. Repeating this procedure 10,000 times produced a distribution of performance changes that reflected the behavioral variability. The p-value of the observed performance change was computed as the fraction of times the bootstrap produced an opposite performance change (e.g. if a performance decrease was observed during photo-stimulation, the p-value was the fraction of times a performance increase was observed during bootstrap, one-tailed test).

**Electrophysiology data analysis.** The extracellular recording traces were band-pass filtered (300-6 kHz). Events that exceeded an amplitude threshold (4 standard deviations of the background) were subjected to manual spike sorting to extract single units[43].

**Stimulus-selective, choice-selective, and lick movement neurons.** Trial types differed in object location ('stimulus', anterior versus posterior), lick direction ('choice', left versus right), and reward ('outcome', rewarded versus unrewarded). We separately computed neuronal selectivity for each variable[46]. For each neuron, we computed the spike counts of individual trials within specific analysis windows. We modeled the neuronal response using a linear model, where the spike count ($R$) is a linear combination of 3 potential contributing variables (i.e. object location, $\alpha$; choice, $\beta$; and outcome, $\gamma$), plus a constant $a$:

$$R = a + (b_1 \cdot \alpha) + (b_2 \cdot \beta) + (b_3 \cdot \gamma) \tag{1}$$

where:

$$\alpha = \begin{cases} 1 \ if \ stimulus \ is \ anterior \\ 0 \ if \ stimulus \ is \ posterior \end{cases} \tag{2}$$

$$\beta = \begin{cases} 1 \ if \ choice \ is \ right \\ 0 \ if \ choice \ is \ left \end{cases} \tag{3}$$

$$\gamma = \begin{cases} 1 \ if \ the \ trial \ is \ rewarded \ (correct \ trial) \\ 0 \ if \ the \ trial \ is \ unrewarded \ (error \ trial) \end{cases} \tag{4}$$

To test if a neuron is selective for stimulus or choice, we tested the significance of each contribution factor using 3-way ANOVA (MATLAB function 'anovan' using 'linear' model) against the null hypotheses that the coefficients $b_1$ or $b_2$ were 0. To classify stimulus- or choice-selective neurons, we used the combined spike counts calculated in both the sample and delay epochs. Testing selectivity using spike counts from either sample or delay epochs yielded similar results. A neuron was deemed stimulus-selective if the null hypothesis $b_1 = 0$ was rejected at $p < 0.01$. A neuron was deemed choice-selective if the null hypothesis $b_2 = 0$ was rejected at $p < 0.01$. A neuron could exhibit significant selectivity for both stimulus and choice if both coefficients were nonzero. Only neurons with enough error trials (5 or more for each trial type) were tested (ALM, 2468 out of 2939 neurons; SC, 621 out of 1147 neurons).

Among stimulus- and choice-selective neurons, we further calculated the stimulus and choice selectivity (Fig. 2 and Supplementary Fig. 3d, f). For each neuron, the spike counts within specific analysis windows (in 200 ms moving windows or during the sample and delay epochs) were grouped according to trial types ("lick right" correct trials, $R_{CR}$; "lick left" correct trials, $R_{CL}$; "lick right" error trials, $R_{ER}$; "lick left" error trials, $R_{EL}$). The selectivity was calculated from the average spike counts ($\langle\rangle$) in each group:

$$Stimulus \ selectivity = \frac{(\langle R_{CR} \rangle - \langle R_{EL} \rangle) + (\langle R_{ER} \rangle - \langle R_{CL} \rangle)}{2} \tag{5}$$

$$Choice \ selectivity = \frac{(\langle R_{CR} \rangle - \langle R_{ER} \rangle) + (\langle R_{EL} \rangle - \langle R_{CL} \rangle)}{2} \tag{6}$$

We separately defined licking movement neurons based on significant firing rate modulation during rhythmic lick cycles. Mice licked at a frequency of 7 Hz[49]. For each lick, spike counts were calculated in two adjacent 50-ms time windows following the detection time of the lick. Across all licks and all trials (correct trials only), neurons with a significant difference in spike count between the two windows were deemed to be modulated by licking ($p < 0.01$, two-tailed t-test). Licking modulation was calculated for each neuron as the mean difference in spike rate between the two windows.

Bootstrap was used to compare the distributions of stimulus/choice-selective neurons to the distribution of licking movement neurons in ALM and SC (Fig. 2e, f). The neuronal dataset was re-sampled 10,000 times with replacement and the distributions of selective neurons were computed on the re-sampled dataset. P value reflected the fraction of times the peak of licking movement neuron distribution was medial to the stimulus-selective neurons or choice-selective neurons (one-tailed test). In addition, we tested whether the distributions of stimulus-selective, choice-selective, and licking movement neurons in ALM and SC deviated significantly from uniform distributions. We permutated the CCF coordinates of the sampled neurons 10,000 times to create null distributions. We then examined whether the peak of the observed distributions deviated significantly from the peaks of null distributions. All distributions significant differed from peaks of null distributions ($p < 0.01$, permutation test).

**Contra- and ipsi-preferring populations.** To examine the representation of contralateral and ipsilateral licking choices, neurons were tested for significant trial-type selectivity using spike counts from "lick left" and "lick right" trials (correct trials only, two-tailed t-test, $p < 0.01$, Fig. 3). Because many neurons were recorded for limited number of error trials, we therefore only used correct trials to calculate selectivity for contralateral vs. ipsilateral choice. Neurons that significantly differentiated "lick left" and "lick right" trials within specific analysis windows were deemed "selective". Selective neurons were further classified into contra-preferring versus ipsi-preferring based on their preferred lick direction relative to the recorded hemisphere (e.g. a contra-preferring neuron from the left hemisphere showed higher spike rate in "lick right" trials). For Fig. 3a, the analysis window was specific epochs of the task. For Fig. 3b, analysis was performed in 200 ms moving windows.

To examine the competitive dynamics of contra-preferring and ipsi-preferring populations, we focused on simultaneously recorded neuronal populations (Fig. 3c-e). In each recording session, we classified neurons into contra-preferring and ipsi-preferring populations based on selectivity computed during the delay epoch. Only sessions with 5 or more selective neurons in each group simultaneously recorded for 30 or more trials were considered. We used a portion of trials (10 trials for each trial type, correct trials only) to define neurons' trial-type preference, we then examined activity dynamics of the sorted populations in independent trials. Many recording sessions showed non-trial-type specific activity drifts over time, such as non-selective ramping during the delay epoch. This produced positive activity correlations between neuron groups as both populations showed this non-selective ramping. To examine the single trial dynamics of contra-preferring versus ipsi-preferring populations free of this global activity drifts over time, we detrended the activity. For each neuron, we calculated its average activity across all trials at each time point and subtracted this average activity. Importantly, the average activity was calculated using the trials that were used to define the contra- and ipsi-preferring populations, independent from the trials used to examine the activity dynamics. This detrending did not artificially introduce anticorrelations between the contra- and ipsi-preferring populations. As a control, we randomly grouped neuron into two populations (i.e. shuffled control) and re-performed the same analysis. Anticorrelated activity was absent between randomly grouped neuronal populations (Fig. 3e and Supplementary Fig. 5b). To quantify the anticorrelated activity of contra- and ipsi-preferring populations in single trials, we averaged the mean-subtracted activity of all contra-preferring or ipsi-preferring neurons at each time point ($\Delta$FR, Fig. 3c-d). We calculated Pearson's correlation between $\Delta$FR of contra-preferring and ipsi-preferring populations across time points. For analysis presented in Fig. 3 and Supplementary Fig. 5, this analysis was performed on activity during the sample and delay epochs. For comparison, we also performed the same analysis on activity in an 800 ms window before the sample epoch and on activity during the response epoch. Anticorrelated activity was absent during these epochs. For SC recordings, this analysis was further limited to the sessions where recording tracks could be reconstructed (Supplementary Fig. 5f).

To examine the co-fluctuation of contra- and ipsi-preferring populations across trials, we calculated their noise correlation within the same trial type (Supplementary Fig. 5c-d). Similar to above, only sessions with 5 or more selective neurons in each group simultaneously recorded for 30 or more trials were considered. A portion of trials was used to define neurons' trial-type preference and independent trials were used to calculate noise correlation. Specifically, we first calculated the average spike rate of each population on individual trials in the last 400 ms window of the delay epoch. For each population, we then calculated its mean spike rate across all trials of the same trial type ("lick left" or "lick right"). This mean spike rate was then

subtracted from the individual trial spike rates of the same trial type, resulting in a ΔFR that reflected the trial-to-trial fluctuation of the population within the same trial type. We then calculated Pearson's correlation between ΔFR's of contra-preferring and ipsi-preferring populations across trials.

**Effects of SC photoactivation and photoinhibition.** To examine the effect of SC photoactivation and photoinhibition on ALM and SC activity (Figs. 4–6), we classified neurons into contra- and ipsi-preferring populations based on selectivity computed during the delay epoch. We used a portion of the control trials (10 trials for each trial type, correct trials only) to define neurons' trial-type preference, we then examined activity of the sorted populations in independent control and photostimulation trials. To compare the activity of control and photostimulation trials (Figs. 4–6), we used both correct and error trials to calculate the average activity in each trial type, where trial types were grouped based on the sensory instruction ("lick contra" and "lick ipsi" relative to the recorded hemisphere). We used both correct and error trials because ALM and SC choice-selective neurons were coupled to upcoming lick direction. If only correct trials were used, the analysis would exclude a significant portion of the trials in which photostimulation caused mice to switch future lick directions, thus underestimating the effects of photostimulation on selectivity.

We used a linear model to distinguish trial-type-specific activity changes induced by SC manipulations from non-specific activity changes across all trial types. We first calculated the activity change between control and photostimulation trials for each neuron in each trial type (ΔFR). The activity change was calculated using the spike rate during the last 200 ms of the delay epoch. To test whether there is a consistent trial-type specific activity change, we fit a linear mixed effect model to with a random effect of recording sessions to account for the activity change. The model was fit to the contra-preferring and ipsi-preferring populations separately. The model has the following form:

$$\Delta FR = \beta_0 + \beta_1(trial\ type) + (\beta_0|session) + (\beta_1(trial\ type)|session) \quad (7)$$

$\beta_0$ is an intercept that captures non-trial-type specific activity changes. $\beta_1$ is a slope that captures activity changes as a function of trial type. Trial type is 0 or 1 for the unaffected and affected trial type respectively (i.e., for SC photoinhibition, lick ipsi trial is 0 and lick contra trial is 1; for SC photoactivation, lick ipsi trial is 1 and lick contra trial is 0). If photostimulation induced non-specific activity changes in both trial types, the activity change would be captured by $\beta_0$, with $\beta_1$ being near 0. On the other hand, if photostimulation selectively induced activity changes only in one trial type, the activity change would be captured by $\beta_1$, and $\beta_0$ would be near 0. To account for variability across sessions, each neuron has a random intercept ($\beta_0|session$) and a random slope ($\beta_1(trial\ type)|session$) for each session. So any session-specific activity changes are absorbed by the random effect parameters and only activity changes common across sessions will be captured by $\beta_0$ and $\beta_1$.

We fit the model using the matlab function *fitlme()*. The results are shown in Supplementary Figs. 6-7. These statistical tests further confirmed our observations: SC photoinhibition induced a significant activity decrease in the contra-preferring population specifically in lick contra trials (i.e., a non-significant $\beta_0$ near 0, and a significantly negative $\beta_1$ slope); SC photoinhibition induced a significant activity increase in the ipsi-preferring population specifically in lick contra trials (i.e., a non-significant $\beta_0$ intercept near 0, and a significantly positive $\beta_1$ slope). SC photoactivation induced the opposite pattern of activity change (Supplementary Fig. 6).

**Activity mode anlaysis.** Across $n$ neurons, we defined a set of orthogonal directions in activity space (**Mode**, $n \times 1$ vectors) that captured components of population activity (Fig. 4d-f). We defined the activity

modes using a portion of the control trials. Separate control trials and photostimulation trials were used for activity projections. At each time point, we calculated the trial-averaged population response vectors (**r**, $n \times 1$) for specific trial types. Activity projections were calculated as $Mode^T r$. To obtain standard errors, we bootstrapped the neurons in the dataset. Standard error was the standard deviation of the activity projections calculated on the resampled datasets.

To capture choice activity of ALM neurons, we found a $n \times 1$ vector (coding dimension, **CD**) in the $n$ dimensional activity space that maximally separates the response vectors in "lick right" trials and "lick left" trials based on the activity during the late delay epoch. To estimate **CD**, we first esimated $CD_t$ at different time points during the delay epoch (in 10 ms steps) using part of control trials (correct trials only). Average spike counts were computed in 400-ms windows in 10-ms steps. For each trial type ("lick right" and "lick left") we computed the average spike counts $\bar{r}_{lick\ right}$ and $\bar{r}_{lick\ left}$, $n \times 1$ response vectors that described the population response at each time point, $t$. $CD_t$ is the difference in the mean response vectors: $CD_t = \bar{r}_{lick\ right} - \bar{r}_{lick\ left}$. During the delay epoch, the direction of $CD_t$ was stable (correlation of $CD_t$'s between the early delay epoch vs. late delay epoch, $0.71 \pm 0.02$, mean ± s.e.m.). We averaged the $CD_t$'s from the last 600 ms of the delay epoch to obtain one **CD**. This fixed **CD** was used for activity projections. The projection along **CD** captured $89.2 \pm 2.6\%$ of the population selectivity for "lick left" and "lick right" trials over the sample and delay epochs (root mean square, RMS, of the spike rate difference between "lick right" trials and "lick left" trials), and $25.2 \pm 4.3\%$ of the total variance in ALM task-related activity. Activity variance was quantified as the RMS of the baseline subtracted activity over the sample and delay epochs.

We additionally defined a ramping mode as $Mode_{ramping} = \bar{r}_{delay} - \bar{r}_{pre\ sample}$, where $\bar{r}_{pre\ sample}$ represents the population response vector 500 ms before the sample epoch and $\bar{r}_{delay}$ represents the population response vector during the last 500 ms of the delay epoch. We further rotated the activity mode using the Gram-Schmidt Process to be fully orthogonal to **CD**. The ramping mode was calculated using the combined responses from correct "lick left" and "lick right" trials. The calculation of the ramping mode followed the procedures in[47,56,84], which by construction captured activity showing a ramp during the delay epoch. Previous analysis in ALM show that activity along this ramping mode correlates with reaction time[56]. The projection along the ramping mode captured <1% of the population selectivity for "lick left" and "lick right" trials over the sample and delay epochs, and $14.7 \pm 2.4\%$ of the total variance in ALM activity over the sample and delay epochs.

Finally, we calculated an activity mode that captured most of the remaining activity variance. We calculated eigenvectors of the population response using singular value decomposition (SVD). The data for the SVD was a $n \times t$ population response matrix containing the baseline-subtracted PSTHs of $n$ neurons during sample and delay epochs ("lick right" and "lick left" trials concatenated). We further rotated the eigenvectors using the Gram-Schmidt Process to be fully orthogonal to **CD** and ramping modes. The eigenvector carrying the most variance was used for activity projections. The projection along this activity mode captured $1.0 \pm 0.2\%$ of the population selectivity for "lick left" and "lick right" trials over the sample and delay epochs, and $51.0 \pm 4.1\%$ of the total variance in ALM activity over the sample and delay epochs.

**Optrode recording and ChR2-tagging analysis.** In VGAT-ChR2-EYFP mice in which we performed SC optrode recordings and SC photoinhibition during the delay epoch, we analyzed the recording data to identify neurons excited and inhibited by photostimulation (Supplementary Fig. 8a-c). Recordings were targeted to the central region of SC where choice-selective neurons were enriched, and where contra-preferring and ipsi-preferring neurons exhibit competitive dynamics.

Electrode tracks were labeled with DiI and recordings outside of the region of interest were not analyzed further. In total, we obtained 225 neurons from the central region of SC out of 474 neurons recorded in 8 mice. To quantify the effect of photostimulation on individual neuron spike rates, we calculated spike counts within the photostimulation window (delay epoch) and compared them to the control trial spike counts in the same time window. Significant spike rate change was tested using two-tailed t-test ($p < 0.01$). "Lick left" and "lick right" trials were pooled. We obtained 36 photoexcited neurons and 95 photo-inhibited neurons (out of 225).

Within the SC neurons excited and inhibited by photostimulation, 15 and 38 respectively exhibited significant trial-type selectivity during the delay epoch. We calculated selectivity for contralateral licking choice (Supplementary Fig. 8b, c). Lick direction was relative to the recorded hemisphere (e.g. "lick right" and "lick left" trials corresponded to lick contra and lick ipsi trials respectively for neurons from the left hemisphere). Contra-selectivity was calculated as the firing rate difference between lick contra and lick ipsi trials for each neuron (only correct trials were included). The firing rate differences were averaged across all selective neurons. Within the SC neurons excited and inhibited by photostimulation, we also quantified the proportion of contra-preferring and ipsi-preferring neurons (Supplementary Fig. 8b-c). Bootstrap was used to evaluate whether contra-preferring or ipsi-preferring neurons were significantly higher in proportion (Supplementary Fig. 8b-c). The neuronal dataset was re-sampled with replacement, and the P value reflected the fraction of times when the opposite preference was observed more frequently (one-tailed test).

Long-duration photostimulation of SC could induce activity changes through long-range pathways with complex temporal dynamics. We additionally used optogenetic tagging to identify GABAergic neurons with 1-ms photostimulation in VGAT-ChR2-EYFP mice or GAD2-ires-cre mice expressing ChR2 in GABAergic neurons. We used a combination of criteria to identify neurons with time-locked responses to photostimulation, including manual inspection of voltage traces (Supplementary Fig. 8d), short response latency (Supplementary Fig. 8e), and significantly higher number of spikes evoked in a 10 ms window following each light pulse compared to baseline spike rate in a 10 ms window before photostimulation (Supplementary Fig. 8e). Significant spike rate change was tested using two-tailed t-test ($p < 0.01$). Response latency was measured at the peak spike rate after the photostimulus onset. From these recordings, we identified 56 GABAergic neurons out of 329 neurons recorded in 7 mice. Within the identified GABAergic neurons, 29 neurons exhibited significant trial-type selectivity during the delay epoch. We calculated their contra-selectivity and the proportion of contra-preferring and ipsi-preferring neurons as above (Fig. 7b).

We used optogenetic tagging to identify SC glutamatergic neurons with 1-ms photostimulation in Vglut2-ires-cre × Ai32 mice. From these recordings, we identified 60 glutamatergic neurons in 2 mice. Within the identified glutamatergic neurons, 41 neurons exhibited significant trial-type selectivity during the delay epoch. We calculated their contra-selectivity and the proportion of contra-preferring and ipsi-preferring neurons as above (Fig. 7c).

**Neural activity prediction from videos of task-performing mice.** To examine if the choice-selective delay activity in ALM and SC could be explained by ongoing movements, we trained convolutional neural networks (CNNs) to predict neural activity from videos of task-performing mice. Due to the limited number of neuronal recording sessions with high quality video recordings, this analysis could not be done for all analysis in the paper. Data from ALM and SC recordings were combined for this analysis. Our goal was to build models that related neural activity to ongoing movements on single trials, then subtract this movement-related activity and examine if any choice selectivity remains in the residual activity (Supplementary Fig. 4e-h).

The CNNs were chosen over linear models because of their superior prediction performance. The details of the CNN model and training procedures have been previously described in detail[61].

The CNNs were trained to predict the activity of individual neurons. The analysis was limited to the neurons with significant selectivity during the delay epoch. The firing rates were computed using 400 ms wide bins with a 100 ms bin stride. The inputs to the CNNs were bottom- and side-view videos of mice during the delay epoch (see *Videography*). The video frames were temporally downsampled by a factor of 10 (down to 20 Hz) and spatially downsampled by a factor of roughly 3 (down to 130 × 104 pixels for the bottom view; 130 × 86 pixels, side view). The CNNs predicted neural activity at each time point from all video frames within a 400 ms time window that was matched to the time bin used to calculate the firing rates. All video frames from the time window were concatenated and fed into the networks. The same CNNs were used for prediction across all time points within the delay epoch.

The CNNs had 6 convolutional layers and 1 fully connected layer shared across sessions (512 units), followed by another fully connected layer specific to each session (128 units). To predict individual neuron activities, 2 additional fully connected layers were used (64 units and 1 final readout unit), and these layers were specific to each neuron in each session. The first seven layers were shared across sessions in order to increase the number of training samples and avoid overfitting. The layers following them were session-specific to account for differences in appearance of mice. The bottom- and side-view video frames were fed into separate 6-layered convolutional networks of identical architecture, whose output activations were concatenated and fed into the session-independent fully connected layer. All convolutional layers had 64 output feature maps and a kernel size of 5 × 5, and a 2 ×2 max-pooling was applied after the third and fifth convolutional layers. Units in each layer had a nonlinear activation function (ReLU, rectified linear units[85]). Batch normalization was applied after each convolutional layer to facilitate training[86]. To reduce overfitting, dropout was applied before the session-independent fully connected layer and the session-dependent fully connected layer with a drop probability of 0.3[87]. The networks were trained by gradient descent to minimize the mean squared error between the predicted and target activity. The CNNs and their training were implemented in Pytorch[88].

The CNNs' prediction performance was evaluated by 5-fold cross-validation (Supplementary Fig. 4g). For each trial type and time point, we computed variance explained ($R^2$) across trials within each trial type:

$$R^2 = 1 - \frac{\sum_{n\,trials}(x_{actual} - x_{predicted})^2}{\sum_{n\,trials}(x_{actual} - \langle x_{actual} \rangle)^2} \tag{8}$$

where $x$ is the spike rate on the $n^{th}$ trial. The $R^2$ was averaged across two trial types and all time points for each selective neuron. Computing $R^2$ across trials was done to ensure that $R^2$ measured how well the networks predicted variability of neural activity across trials, rather than an average change of neural activity across trial types or time.

### Anatomical data analysis
**Alignment of electrophysiology recording into CCF.** Recording locations were recovered *post-hoc* by identifying coronal sections containing DiI labeled recording tracks. Electrodes were manually placed along the DiI labeled tracks. Electrode locations were further adjusted along the track based on manipulator readings and the lamination of activity patterns across the electrodes, which corresponded well to anatomical structures: high activity levels in neocortex and SC, low activity levels between them. Single-unit locations were determined based on the locations of the electrodes the units were recorded on. Finally, we aligned the coronal sections (and by

extension, unit locations within them) to the Allen Mouse Common Coordinate Framework (CCF)[83] using landmark-based image registration (Fig. 2b)[52]. The registration target was the 10 μm/voxel CCF anatomical template brain. To align a coronal section, we first selected the coronal plane in the anatomical template that best corresponded to the section. Next, we manually placed control points at corresponding anatomical landmarks in each image. 30-50 landmarks were selected in a single image. Next, the image was warped to the CCF using an affine transformation followed by a nonrigid transformation using b-splines[89]. Images were warped using the B-spline Grid, Image and Point based Registration package available on the Matlab FileExchange (https://www.mathworks.com/matlabcentral/fileexchange/20057-b-spline-grid--image-and-point-based-registration).

**Anterograde anatomical tracing of ALM and SC connectivity.** For brains containing anterograde tracers injected in ALM or SC, we obtained whole-brain 3D image volumes made up of 50 μm coronal sections. Each coronal section was made up of 80–200 tiles merged with Neurolucida software. The whole-brain 3D volume was warped into the CCF using a Matlab-based script[47] similar to the one used for the alignment of 2D coronal sections described above (Supplementary Fig. 1b). Anatomical landmark correspondences between the whole-brain 3D volume and the 10 μm/voxel CCF anatomical template brain were manually annotated. A 3D volume typically requires 200–300 landmarks to define an accurate transformation.

We quantified the descending projections of ALM and SC in the medulla based on anterograde fluorescence intensity (Supplementary Fig. 1f-h). Quantifications of fluorescence were performed on images postalignment to the CCF. We found consistent labeling patterns in the medulla across different injection cases (Supplementary Fig. 1f-h). Alignment to the CCF allowed us to quantify fluorescence overlaps across different injection cases. To quantify the overlap, we thresholded the fluorescence intensity at 0.3 of the maximum intensity (see an example thresholded image in Supplementary Fig. 1c). The labeled area was defined as all pixels that exceeded this threshold (Supplementary Fig. 1f-h). The overlap between ALM and SC descending projections in the medulla was calculated as the number of pixels co-labeled by ALM injections and SC injections (Supplementary Fig. 1h).

We also quantified the connectivity between ALM and SC in the thalamus based on the overlap of anterograde fluorescence from SC and ALM-projecting thalamus (Fig. 1d and Supplementary Fig. 1j-l). Quantifications of the fluorescence overlap were always made in brains that contained co-injections of anterograde tracers in SC and retrograde tracers in ALM. Fluorescence overlaps were calculated in the same way as described above (Supplementary Fig. 1l).

We quantified the topography of ALM descending projections in SC and thalamus based on anterograde fluorescences from dual injections in the medial and lateral ALM (Supplementary Fig. 1d-e). Quantifications of fluorescence were performed on images post alignment to the CCF. Allen Reference Atlas annotation of SC and thalamus were used to only analyze fluorescence within the brain area of interest. The fluorescence intensity of all cases were averaged to obtain the fluorescence profiles in Fig. 2g, h.

**Retrograde tracing of SC medulla-projecting and thalamus-projecting neurons.** For brains containing retrograde tracers injected in IRt and VM, we obtained whole-brain image volumes as described above. In coronal sections covering the rostral to caudal extend of SC, we manually annotated labeled neurons in ImageJ (Supplementary Fig. 1i). Next, the whole-brain 3D volume (and annotated neuron locations within it) was warped into the CCF. The distribution of the annotated neurons was quantified in the CCF (Fig. 4i).

## Statistics

The sample sizes were similar to sample sizes used in the field: for behavior, 3 mice or more per condition. No statistical methods were used to determine sample size. All key results were replicated in multiple mice. Mice were allocated into experimental groups according to their strain. Unless stated otherwise, the investigators were not blinded to allocation during experiments and outcome assessment. Trial types were randomly determined by a computer program. During spike sorting, experimenters cannot tell the trial type, so experimenters were blind to conditions. Statistical comparisons using t-tests, bootstrap, and other statistical tests are described in detail in the sections above.

### Reporting summary

Further information on research design is available in the Nature Portfolio Reporting Summary linked to this article.

## Data availability

Processed data and source data to reproduce the figures are avaliable on Zenodo at https://zenodo.org/records/8141357. Source data are provided with this paper.

## Code availability

Code used for data analysis is available at https://github.com/NuoLiLabBCM/ThomasYangEtAL2023NC.

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

## Acknowledgements

We thank Z Guo, JM Yau, A Finkelstein, B Kang, JH Kim, RB Dewell, E Yttri for comments on the manuscript and insightful discussions. This work was funded by the Robert and Janice McNair Foundation (N.L.), Whitehall Foundation (N.L.), Alfred P. Sloan Foundation (N.L.), Searle Scholars Program (N.L.), Pew Scholars Program, NIH NS112312 (A.T. and N.L.), NS113110 (N.L.), NS131229 (N.L.), K01 NS119372 (A.T.), NIMH IRP ZIA MH002497-34 (C.G.), McKnight Foundation (N.L.), and Simons Collaboration on the Global Brain (N.L.).

## Author contributions

A.T. and N.L. conceived and designed the experiments. A.T. and W.Y. performed electrophysiology and behavioral experiments with help from S.L.T., B.S., K.S. and M.T. G.C. and W.Y. contributed electrophysiology data from ALM. A.T., C.G. and W.Y. performed anatomical experiments. C.W. performed video analysis. A.T., W.Y. and N.L. analyzed data. A.T. and N.L. wrote the paper with inputs from all authors.

## Competing interests

The author declares no competing interests.
