## [Peer Review File · Nature Communications]

Superior colliculus bidirectionally modulates choice activity in frontal cortexEditorial Note: This manuscript has been previously reviewed at another journal that is not operating a transparent peer review scheme. This document only contains reviewer comments and rebuttal letters for versions considered at *Nature Communications*.

REVIEWER COMMENTS

Reviewer #1 (Remarks to the Author):

All my concerns have been addressed through the revision process, and I support the publication of this manuscript.

Reviewer #2 (Remarks to the Author):

We believe the authors have addressed our concerns and we have no further comments.

Reviewer #3 (Remarks to the Author):

In the current version of the manuscript the authors show that a topographically registered path exists from the ALM to the SC and further to the medulla, forming a descending motor pathway for directional licking. The authors also demonstrate that populations in the SC and ALM encode opposing choices. Further, GABAergic and glutamatergic SC neurons also encode opposing action choices. The authors show that transiently and unilaterally inactivating SC activity, in a cell type specific way, can bidirectionally modulate ALM activity and behavior, thus suggesting that SC is the driver of action choice as well as motor choice related activity in the frontal cortex.

We have previously reviewed this manuscript. Most of our previous suggestions have been incorporated by the authors. The current manuscript is a significant improvement over the previous submissions. Overall, we applaud the authors for generating this enormous and rich data set using several neuroscientific techniques. The paper now has a reasonably cohesive story and interpretations. However, some major concerns still persist which we have highlighted below and which, in our opinion, need to be addressed.

Major points.

1. Lines 20-24 ☐ “SC GABAergic neurons encoded ipsilateral choice and glutamatergic neurons encoded contralateral choice, and activating or suppressing these cell types could bidirectionally drive push-pull choice activity in frontal cortex”

Lines 549-552 ☐ “Our findings suggest a circuit mechanism whereby SC GABAergic neurons encode ipsilateral choice and glutamatergic neurons encode contralateral choice during action selection, and the opposing actions of these SC cell types opposingly drive push-pull choice competition activity across the action selection network”

These strong statements have been a persistent issue we have had with the interpretation that the authors propose. While the results shown do unequivocally demonstrate that SC GABAergic and glutamatergic neurons encode opposing choices, an important point that the authors never mention that these neurons are directly connected (Kasai et al 2016). Indeed, the authors themselves exploit this fact when they activate SCid inhibitory neurons to silence glutamatergic neurons (e.g., Fig. 4, line 259-260 – photoinhibition of SC by activating the excitatory opsin, ChR2, in VGAT+ neurons.).

The authors statements throughout the manuscript seem to suggest that these populations are independent of each other and that they form the ‘decision’ of encoding opposite choice via separate mechanisms. However, the fact very well might be that the reason glutamatergic neurons encode contralateral choice is because GABAergic neurons which connect with them are encoding the opposing choice. Thus, if these contra-choice encoding glutamatergic neurons further project to the ALM (currently there is no evidence that SC GABAergic neurons send projections to the ALM), then bidirectionally manipulating SC GABAergic neuron activity should opposingly manipulate ALM activity.

Thus, we feel that the claim, “because SC GABAergic and glutamatergic neurons encode opposing choice, this is a discovery of a circuit mechanism”, potentially misleading.

Rather, we suggest explicitly mentioning the possibility that these SC glutamatergic and GABAergic neurons form an expected direct functional circuit where encoding opposing choice is the only logical outcome. Additionally, we recommend appropriately revising the sentence in the abstract.

2. Push-pull activity within SC cell types

Lines 207 ☐ “Models of action selection generally invoke pools of neurons representing competing choice options, and actions are selected through competition between these neuronal populations”

This point is again related to point (1), and in a sense more severe. We suggest that in the discussion, the authors explicitly mention that a strong possibility is as depicted in Figure 5, where the ipsi encoding GABAergic and contra encoding glutamatergic neurons, form a common pool. And, that the real competition is between the hemispheres. Without evidence to the contrary, promoting the “model” of competition or between the GABAergic and glutamatergic pools is misleading and incorrect.

We further elaborate our points below.

a. The authors cite important modelling papers that propose that pools of neurons representing different choices, compete with each other.

However, the authors miss the fact that in each of these papers (cited by the authors), within each pool there are populations of both excitatory and inhibitory neurons, and that these populations (pools) non-trivially inhibit each other via feedback inhibition, which is necessary for competition to be resolved in a categorical and flexible manner.

b. In the manuscript, the authors seem to indicate that SC GABAergic and glutamatergic populations are the populations which represent the pools of neurons encoding opposing choices.

This seems to be a misinterpretation of the popular mutual inhibition models (Machens et al, 2005, Wang et al, 2002 etc, also cited by the authors).

3. Unilateral inactivation of ALM does not disrupt behavior and also does not cause a sustained disruption of delay period activity in the SC. (lines #381 to 397)

While the authors have satisfactorily addressed the concerns regarding this point that we raised in their previous submission, we feel that there needs to be an explicit statement in the discussion to the effect that “Bilateral inactivation of ALM may very well disrupt behavior as well as delay period activity in the SC, consistent with our previous work (Li et al 2016, Nature), and is an open question.” We believe such a statement is necessary to avoid potential misconceptions about the descending contribution of ALM activity to SC activity, given their previous work.

4. Subtraction of movement related activity (lines 200 and lines 1840-1850)

“To eliminate the possibility that choice selective activity in the early sampling phase may be attributable to ongoing movements of the mice, which could differ in ‘lick left’ and ‘lick right’...”

a. The approach of subtracting CNN-predicted ALM and SC activity from recorded neural activity first requires validation of the approach. In other words, can we “trust” that the CNN output is as a reliable estimate of the “truth”? Some validation approach is essential, or alternatively, a discussion of why validation is not needed is essential. One possibility is to test the performance of the CNN during the opto-inactivation trials during the Sample epoch and show that it performs well. Considering that one of the findings is that the early choice related activity (during the Sample phase) was explainable by the orofacial movements, we would expect inactivation of the SC during the Sample phase to suppress this activity.

b. The authors indicate that they subtract this predicted activity from the actual activity.

It is unclear if this was done for all the figures in the paper or only the Extended figure 5h. This is important, because although this approach is interesting, the results are still correlational and simply removing this would be detrimental to the interpretation of subsequent results.

c. It is also unclear from the methods how the 3D volume of space around the mouse head was calibrated and the precision of the calibration is also not provided.

MINOR POINTS

1. Line #260 – “VGAT-ChR2-EYFP mice” . For clarity can you mention clearly that this is a crossed transgenic line between VGAT-cre and ... (just pointing to the Methods the first time this term is used will already be sufficient)

REVIEWER COMMENTS

Reviewer #1 (Remarks to the Author):

All my concerns have been addressed through the revision process, and I support the publication of this manuscript.

We thank the reviewer for the constructive review that improved the paper.

Reviewer #2 (Remarks to the Author):

We believe the authors have addressed our concerns and we have no further comments.

We thank the reviewer for the constructive review that improved the paper.

Reviewer #3 (Remarks to the Author):

In the current version of the manuscript the authors show that a topographically registered path exists from the ALM to the SC and further to the medulla, forming a descending motor pathway for directional licking. The authors also demonstrate that populations in the SC and ALM encode opposing choices. Further, GABAergic and glutamatergic SC neurons also encode opposing action choices. The authors show that transiently and unilaterally inactivating SC activity, in a cell type specific way, can bidirectionally modulate ALM activity and behavior, thus suggesting that SC is the driver of action choice as well as motor choice related activity in the frontal cortex.

We have previously reviewed this manuscript. Most of our previous suggestions have been incorporated by the authors. The current manuscript is a significant improvement over the previous submissions. Overall, we applaud the authors for generating this enormous and rich data set using several neuroscientific techniques. The paper now has a reasonably cohesive story and interpretations. However, some major concerns still persist which we have highlighted below and which, in our opinion, need to be addressed.

We thank Reviewer #3 for their support of our work. We address the remaining comments below.

Major points.

1. Lines 20-24 ◊ “SC GABAergic neurons encoded ipsilateral choice and glutamatergic neurons encoded contralateral choice, and activating or suppressing these cell types could bidirectionally drive push-pull choice activity in frontal cortex”

Lines 549-552 ◊ “Our findings suggest a circuit mechanism whereby SC GABAergic neurons encode ipsilateral choice and glutamatergic neurons encode contralateral choice during action

selection, and the opposing actions of these SC cell types opposingly drive push-pull choice competition activity across the action selection network”

These strong statements have been a persistent issue we have had with the interpretation that the authors propose. While the results shown do unequivocally demonstrate that SC GABAergic and glutamergic neurons encode opposing choices, an important point that the authors never mention that these neurons are directly connected (Kasai et al 2016). Indeed, the authors themselves exploit this fact when they activate SCid inhibitory neurons to silence glutamergic neurons (e.g., Fig. 4, line 259-260 – photoinhibition of SC by activating the excitatory opsin, ChR2, in VGAT+ neurons.).

The authors statements throughout the manuscript seem to suggest that these populations are independent of each other and that they form the ‘decision’ of encoding opposite choice via separate mechanisms. However, the fact very well might be that the reason glutamergic neurons encode contralateral choice is because GABAergic neurons which connect with them are encoding the opposing choice. Thus, if these contra-choice encoding glutamergic neurons further project to the ALM (currently there is no evidence that SC GABAergic neurons send projections to the ALM), then bidirectionally manipulating SC GABAergic neuron activity should opposingly manipulate ALM activity.

Thus, we feel that the claim, “because SC GABAergic and glutamergic neurons encode opposing choice, this is a discovery of a circuit mechanism”, potentially misleading.

Rather, we suggest explicitly mentioning the possibility that these SC glutamergic and GABAergic neurons form an expected direct functional circuit where encoding opposing choice is the only logical outcome. Additionally, we recommend appropriately revising the sentence in the abstract.

We agree with the reviewer that the two SC cell types are not independent, exactly for the reasons stated by the reviewer. It is not our intention to overlook local SC circuit connectivity and it likely directly contributes to the opposing choice encoding of the SC glutamergic and GABAergic populations. We now explicitly mention the possibility that these cell types form a functional circuit.

Previous lines 20-24 now reads “*SC GABAergic neurons encoded ipsilateral choice and locally inhibited glutamergic neurons that encoded contralateral choice. Activating or suppressing these cell types could bidirectionally drive push-pull choice activity in frontal cortex.*”

Previous lines 549-552 now reads “*SC GABAergic neurons locally inhibit glutamergic neurons^{65,66}. Our findings suggest that these SC cell types form a functional circuit during action selection whereby SC GABAergic neurons encode ipsilateral choice and inhibit glutamergic neurons which encode contralateral choice. The glutamergic neurons further project to ALM via thalamus (Fig 7f). Modulating these SC cell types can bidirectionally drive push-pull choice competition activity across the action selection network. The opposing actions of SC cell types likely reflect their local inhibitory connectivity.*”

We have gone through the entire paper to revise the text accordingly to avoid stating these cell types act independently, rather we state that they form a circuit. See manuscript with tracked changes.

We disagree that this functional circuit is expected and that SC cell types encoding opposing choice is the only logical outcome given this circuit connectivity. While SC GABAergic neurons local inhibit glutamatergic neurons, it does not follow that the two cell types will always exhibit opposite encodings. In fact, the two cell types can exhibit the same encoding. We found that SC GABAergic and glutamatergic neurons both exhibit contralateral preference during the motor response. The opposing choice encodings only occur during the delay epoch (Supplementary Fig. 9). The opposing choice encodings likely reflect circuit dynamics specifically related to action selection. We have now highlighted this point in the discussion:

“Intriguingly, the opposing choice encoding of SC cell types is only observed during the delay epoch whereas during the response epoch both cell types exhibit congruent preference for contralateral choice (Supplementary Fig. 9). Manipulating the SC GABAergic neurons during the response epoch also does not produce bidirectional effects on choice, in contrast to the delay epoch (compare Supplementary Figs. 2d and 9f). More work is needed to understand how choice-related activity impinges upon SC cell-types and how these neural populations interact across action selection and motor response phases of the behavior.”

2. Push-pull activity within SC cell types

Lines 207 ◊ “Models of action selection generally invoke pools of neurons representing competing choice options, and actions are selected through competition between these neuronal populations”

This point is again related to point (1), and in a sense more severe. We suggest that in the discussion, the authors explicitly mention that a strong possibility is as depicted in Figure 5, where the ipsi encoding GABAergic and contra encoding glutamatergic neurons, form a common pool. And, that the real competition is between the hemispheres. Without evidence to the contrary, promoting the “model” of competition or between the GABAergic and glutamatergic pools is misleading and incorrect.

We have revised the discussion to explicitly mention that competition between choice options likely occurs across hemispheres. Our data supports cross-hemisphere inhibition: activating SC GABAergic neurons locally inhibit contra-preferring neurons within the same hemisphere and disinhibits contra-preferring neurons in the other hemisphere.

“Competition between choice options likely occurs across hemispheres. Activating SC GABAergic neurons locally inhibit contra-preferring neurons within the same hemisphere (Fig. 7f) and disinhibits contra-preferring neurons in the other hemisphere (Fig. 5), which suggests interhemispheric mutual inhibition that coordinates choice competition in both hemispheres^{33,60}.”

However, our data additionally provide evidence for a local mechanism within-hemisphere during action selection. Our cell-type specific recordings show that GABAergic and glutamatergic neurons within the same hemisphere encode opposite selectivity and modulating each of these cell types during the delay epoch can bidirectionally bias future choice. Again, this contrasts with the motor response epoch, in which both cell types in each SC hemisphere primarily exhibit contralateral encodings, consistent with only cross-hemisphere inhibition.

Manipulating the SC GABAergic neurons during the response epoch also does not produce bidirectional effect on choice, in contrast to the delay epoch (compare delay vs. response epoch in Supplementary Figs. 2d and 9f). The opposing choice encodings during the delay epoch thus reveal a circuit dynamic specifically related to action selection.

We further elaborate our points below.

a. The authors cite important modelling papers that propose that pools of neurons representing different choices, compete with each other.

However, the authors miss the fact that in each of these papers (cited by the authors), within each pool there are populations of both excitatory and inhibitory neurons, and that these populations (pools) non-trivially inhibit each other via feedback inhibition, which is necessary for competition to be resolved in a categorical and flexible manner.

b. In the manuscript, the authors seem to indicate that SC GABAergic and glutamatergic populations are the populations which represent the pools of neurons encoding opposing choices.

This seems to be a misinterpretation of the popular mutual inhibition models (Machens et al, 2005, Wang et al, 2002 etc, also cited by the authors).

We appreciate the concerns raised by the reviewer on this point. We have removed line 207.

Both ALM and SC exhibit pools of neurons encoding opposite choice and show push-pull. Our results show that SC circuits can drive this push-pull in ALM.

In addition, we previously cite these models in the first paragraph of the introduction where we introduce the notion of choice competition. We now write the following in the introduction when summarizing our results around choice competition, which now also highlights the competition between two SC hemispheres.

“Cell-type specific recordings and manipulations in SC further revealed a circuit mechanism for choice competition: within each SC hemisphere, GABAergic neurons encode ipsilateral choice and inhibit glutamatergic neurons encoding contralateral choice, and the two SC hemispheres mutually inhibit each other, which collectively drive push-pull between contra-preferring and ipsi-preferring populations in ALM.”

3. Unilateral inactivation of ALM does not disrupt behavior and also does not cause a sustained disruption of delay period activity in the SC. (lines #381 to 397)

While the authors have satisfactorily addressed the concerns regarding this point that we raised in their previous submission, we feel that there needs to be an explicit statement in the discussion to the effect that “Bilateral inactivation of ALM may very well disrupt behavior as well as delay period activity in the SC, consistent with our previous work (Li et al 2016, Nature), and is an open question.” We believe such a statement is necessary to avoid potential misconceptions about the descending contribution of ALM activity to SC activity, given their previous work.

We have added this statement to the discussion.

4. Subtraction of movement related activity (lines 200 and lines 1840-1850)

“To eliminate the possibility that choice selective activity in the early sampling phase may be attributable to ongoing movements of the mice, which could differ in ‘lick left’ and ‘lick right’...”

The original text in the paper reads “*We considered the possibility that choice selective activity during the delay epoch might be attributable to mice’s ongoing movements, which could differ in “lick left” and “lick right” trials.*”

We clarify that this video analysis focuses on delay epoch activity, not sample epoch activity.

a. The approach of subtracting CNN-predicted ALM and SC activity from recorded neural activity first requires validation of the approach. In other words, can we “trust” that the CNN output is as a reliable estimate of the “truth”? Some validation approach is essential, or alternatively, a discussion of why validation is not needed is essential. One possibility is to test the performance of the CNN during the opto-inactivation trials during the Sample epoch and show that it performs well. Considering that one of the findings is that the early choice related activity (during the Sample phase) was explainable by the orofacial movements, we would expect inactivation of the SC during the Sample phase to suppress this activity.

We thank the reviewer for this point, but we are unsure what the reviewer meant by “test the performance of the CNN during the opto-inactivation trials during the Sample epoch and show that it performs well”. The CNN video analysis in Supplementary Fig. 4f applies only to control trials and it is meant to examine whether ongoing movements could explain ALM and SC choice selectivity during the delay epoch. See main text on page 6:

“We considered the possibility that choice selective activity during the delay epoch might be attributable to mice’s ongoing movements, which could differ in “lick left” and “lick right” trials. To address this, we built convolutional neural networks (CNN) to predict neurons’ firing rate from videos of orofacial movements (Methods). The model predicted a significant portion of ALM and SC activity on single trials (Supplementary Fig. 4e-g). We then subtracted this movement-related activity from ALM and SC activity and the choice selectivity remained in the residuals (Supplementary Fig. 4h). This video analysis shows that ongoing movements could not explain the choice selectivity during the delay epoch.”

We previously validated the approach by calculating cross-validated R^2 of how well CNN could predict neural activity from movement (Supplementary Fig. 4g). CNN predicted activity could explain a substantial proportion of ALM and SC activity variance across trials (30%). The cross-validated R^2 in Supplementary Fig. 4g is calculated across all task epochs. Here we show the R^2 values broken out for each task epoch (Figure R1). R^2 is generally positive across all 3 epochs, indicating that CNN is able to predict a significant portion of the activity from movement across all 3 epochs. Notably, CNN predicted the most activity variance during the response epoch. This makes sense given that movement is most pronounced during the response epoch, which indicates that CNN can predict movement-related activity.

b. The authors indicate that they subtract this predicted activity from the actual activity. It is unclear if this was done for all the figures in the paper or only

Figure R1. Cross-validated R^2 of activity prediction. Bar, mean; circles, individual neurons ($n = 92$). Same as Supplemental Fig. 4g.

the Extended figure 5h. This is important, because although this approach is interesting, the results are still correlational and simply removing this would be detrimental to the interpretation of subsequent results.

The reviewer might be referring to the video analysis in Supplementary Fig 4h (instead 5h). This analysis was only performed on Supplementary Fig. 4h. As explained in the previous round of rebuttal, due to a limited number of neuronal recording sessions with high-quality video recording, this analysis could not be done for all analysis in the paper. This was indicated in the legend of Supplementary Fig 4 and Methods. We have now more clearly indicated this in the Methods. Nevertheless, the results in Supplementary Fig. 4h show that ongoing movements could not explain the choice selectivity during the delay epoch.

c. It is also unclear from the methods how the 3D volume of space around the mouse head was calibrated and the precision of the calibration is also not provided.

We are unsure what the reviewer meant by “the 3D volume of space around the mouse head” and what “calibration” meant. The mouse’s head occupies the majority of the space in the video (see example frames in Supplementary Fig. 4e). As explained in the previous round of rebuttal, the frame-to-frame intensity differences in the pixels surrounding the mouse is near zero due to the surrounding being stationary. To illustrate that these regions carry no predictive information, here we tested the CNN models on videos in which we cropped out the mouse’s head (Figure R2). The predictive power (cross-validated R^2) drops to below zero, i.e. no activity-predicting information in the pixels surrounding the mouse. Note that cross-validated R^2 can be below zero if the model overfits and does not generalize (see R^2 calculation in Methods).

MINOR POINTS

1. Line #260 – “VGAT-ChR2-EYFP mice” . For clarity can you mention clearly that this is a crossed transgenic line between VGAT-cre and ... (just pointing to the Methods the first time this term is used will already be sufficient)

We have made these changes to the main text and the methods section.

REVIEWERS' COMMENTS

Reviewer #3 (Remarks to the Author):

The authors have addressed our comments and we support the publication of this manuscript. Kudos to the authors for the difficult experiments and extensive dataset.

Two last points:

1) The authors use the phrase 'push-pull' at several places in their manuscript to describe their results. By definition, push-pull implies the presence of independent push (excitatory) as well as pull (inhibitory) modulation a circuit; here, ALM - say, the contra preferring population. Direct evidence for push-pull has not been provided. Rather, results in the paper (Figure 5) can be realized using just a mutual-inhibition circuit, which is a "pull-pull" circuit with stimulus input providing excitatory drive. (Separately, the finding that activation vs. silencing of upstream SC neurons causes opposing effects in ALM us not evidence for push-pull.) For clarity, we recommend use of the more transparent phrase, 'competitive interactions' instead, throughout the ms (including, lines 60-65: "Cell-type specific recordings...") - one that does not take anything away from their findings/conclusions.

2) Similarly, to improve clarity, we recommend a slight change to the title. A recommended alternative is: "Superior colliculus bidirectionally modulates choice activity in frontal cortex". This more clearly highlights the central advance of their ms, and takes nothing away from the finding of opposite preferences for gaba-ergic vs glu-ergic cells in SC.

We do not need to see the ms again.

REVIEWER COMMENTS

Reviewer #3 (Remarks to the Author):

The authors have addressed our comments and we support the publication of this manuscript. Kudos to the authors for the difficult experiments and extensive dataset.

Two last points:

1) The authors use the phrase 'push-pull' at several places in their manuscript to describe their results. By definition, push-pull implies the presence of independent push (excitatory) as well as pull (inhibitory) modulation a circuit; here, ALM - say, the contra preferring population. Direct evidence for push-pull has not been provided. Rather, results in the paper (Figure 5) can be realized using just a mutual-inhibition circuit, which is a "pull-pull" circuit with stimulus input providing excitatory drive. (Separately, the finding that activation vs. silencing of upstream SC neurons causes opposing effects in ALM is not evidence for push-pull.) For clarity, we recommend use of the more transparent phrase, 'competitive interactions' instead, throughout the ms (including, lines 60-65: "Cell-type specific recordings...") - one that does not take anything away from their findings/conclusions.

We agree with the reviewer that our findings support a mutual inhibition circuit within SC between the hemispheres, as outlined in Figure 5. We have made the suggested change in the description around Figure 5, in line 60-65, as well as in other places of the text describing circuit interactions within SC.

We note that in certain places of the paper, "push-pull" refers to single trial dynamics between contra- and ipsi-preferring populations in ALM, which are excitatory populations encoding opposing choice and their competitive dynamics are widely deemed as push-pull, and this motivates the current study (as described in the first paragraph of the introduction). We therefore kept the term in specific places of the text describing ALM dynamics. In addition, we feel the SC manipulation experiments in Figure 4 provide sufficient evidence for SC biasing this push-pull in ALM. Namely, we show that activating SC excites ALM contra-population ("push") and inhibiting SC suppresses ALM contra-population ("pull").

2) Similarly, to improve clarity, we recommend a slight change to the title. A recommended alternative is: "Superior colliculus bidirectionally modulates choice activity in frontal cortex". This more clearly highlights the central advance of their ms, and takes nothing away from the finding of opposite preferences for gaba-ergic vs glu-ergic cells in SC.

We have made this change to the title.

We do not need to see the ms again.

We thank the reviewers for their comments and their support for publication.